# Abstraction for Bayesian Reinforcement Learning in Factored POMDPs

**Rolf A. N. Starre**                                                      *r.a.n.starre@tudelft.nl*
*Delft University of Technology*

**Sammie Katt**                                                      *sammie.katt@aalto.fi*
*Aalto University*

**Mustafa Mert Çelikok**                                            *m.m.celikok@tudelft.nl*
*Delft University of Technology*

**Marco Loog**                                                      *marco.loog@ru.nl*
*Radbout University*

**Frans A. Oliehoek**                                              *f.a.oliehoek@tudelft.nl*
*Delft University of Technology*

**Reviewed on OpenReview:** *https://openreview.net/forum?id=HHgdT6m9L9*

## Abstract

Bayesian reinforcement learning provides an elegant solution to addressing the exploration–exploitation trade-off in Partially Observable Markov Decision Processes (POMDPs) when the environment's dynamics and reward function are initially unknown. By maintaining a belief over these unknown components and the state, the agent can effectively learn the environment's dynamics and optimize their policy. However, scaling Bayesian reinforcement learning methods to large problems remains to be a significant challenge. While prior work has leveraged factored models and online sample-based planning to address this issue, these approaches often retain unnecessarily complex models and factors within the belief space that have minimal impact on the optimal policy. While this complexity might be necessary for accurate model learning, in reinforcement learning, the primary objective is not to recover the ground truth model but to optimize the policy for maximizing the expected sum of rewards. Abstraction offers a way to reduce model complexity by removing factors that are less relevant to achieving high rewards. In this work, we propose and analyze the integration of abstraction with online planning in factored POMDPs. Our empirical results demonstrate two key benefits. First, abstraction reduces model size, enabling faster simulations and thus more planning simulations within a fixed runtime. Second, abstraction enhances performance even with a fixed number of simulations due to greater statistical strength. These results underscore the potential of abstraction to improve both the scalability and effectiveness of Bayesian reinforcement learning in factored POMDPs.

## 1 Introduction

Deep reinforcement learning methods have achieved significant milestones, such as attaining superhuman performance on Atari games with only 100k frames (Ye et al., 2021), solving highly complex games such as Go (Silver et al., 2016), and achieving high performance in simulated control tasks (Haarnoja et al., 2018). Most of these achievements rely on recent function approximation advances made with the work on deep neural networks. However, despite these advances, Reinforcement Learning (RL) still faces critical hurdles that must be addressed to enable its application in diverse real-world scenarios. One of the most pressing

challenges is the high sample complexity of deep RL methods, which remains problematic in real-world applications where data collection is expensive, difficult, or dangerous. Fortunately, many such applications offer prior knowledge that can be leveraged to reduce sample complexity. To effectively utilize this knowledge, it is crucial to move away from the tabula rasa approaches of neural networks and incorporate domain-specific prior knowledge into the learning process (Jonschkowski & Brock, 2015; De Bruin et al., 2018; Katt et al., 2022).

Another critical challenge is exploration, which is strongly tied to sample complexity. Effective exploration of unknown and interesting parts of the environment is essential in almost every application. A better exploration strategy means faster learning and improved sample efficiency. Importantly, in RL, an agent must balance exploration (i.e., learning) with exploitation (i.e., maximizing reward). Most deep RL methods rely on heuristics for exploration, which can perform poorly in complex domains (Osband et al., 2016). These challenges are particularly pronounced in partially observable environments, where agents must make decisions based on limited information.

Model-based Bayesian RL (BRL) (Ross et al., 2011) offers a principled approach to addressing the exploration-exploitation trade-off by maintaining a belief over the environment's state and dynamics. This belief enables the agent to balance the exploration–exploitation trade-off effectively. In this work, we build on the Factored Bayes-Adaptive POMDP (FBA-POMDP) framework (Katt et al., 2017; 2019), a model-based BRL approach that combines partial observability and structured factored models. FBA-POMDPs incorporate factorized representations of the environment's dynamics, allowing agents to exploit problem structure for improved scalability. Additionally, thanks to its Bayesian nature, prior knowledge can be incorporated into FBA-POMDPs in a principled way via Bayesian priors, further improving sample efficiency.

Despite its advantages, the FBA-POMDP framework faces significant challenges. While factorization enables better generalization, the inclusion of irrelevant state factors in the model can lead to unnecessarily large model spaces. This increases computational demands during planning and reduces statistical strength by hypothesizing dependencies that are irrelevant to maximizing rewards. For instance, in a cluttered environment, an agent may only need to consider the positions of objects to navigate effectively, while features such as their colors or shapes are irrelevant for the reward. Abstracting away such unnecessary details can simplify the model space, improve computational efficiency, and enhance learning performance. Previous studies have demonstrated that even lossy abstractions can improve performance in planning (He et al., 2020; Chitnis et al., 2021). This is because simplified models can generate more simulations within a fixed runtime, potentially leading to better results in sampling-based online planning. Motivated by this insight, we propose incorporating abstraction into the FBA-POMDP framework to improve scalability and learning efficiency.

We focus on discrete Factored POMDPs (F-POMDPs) and explore the application of abstraction within the FBA-POMDPs framework to enhance planning efficiency, scalability, and learning performance. To achieve this, we augment Factored Bayes-Adaptive Partially Observable Monte-Carlo Planning (FBA-POMCP) (Katt et al., 2017; 2019), an established online planning and learning algorithm for FBA-POMDPs, to incorporate multiple levels of abstraction. This contribution addresses abstraction discovery in the partially observable reinforcement learning setting, where the lack of access to the true model and the complexity of dealing with partial observability make the creation and utilization of abstractions both important and challenging.

We base ourselves on a previous abstraction method for *planning* in fully observable factored Markov decision processs (MDPs) (Dearden & Boutilier, 1997). It creates abstractions automatically based on the problem's structure, enabling agents to plan and learn more effectively. Our approach extends this idea to a partially observable *learning* setting, where the underlying dynamics are not known beforehand, introducing additional complexities. In this setting, the combination of an unknown model, partial observability, and inherent inexactness of approximate abstractions makes it difficult to evaluate abstractions in advance, making the ability to dynamically update abstract models during learning crucial. To address this challenge, we propose to automatically construct abstractions based on the hypothesized connectivity of state factors to the reward function and leverage the Bayes-adaptive framework to adapt the abstract model over time. A key feature

of our approach is its ability to maintain a belief over multiple candidate structures and their corresponding abstract models, ensuring consistent belief updates of the (abstract) models.

Our work represents a novel step toward combining abstraction with BRL in F-POMDPs. Empirically, we demonstrate that abstraction improves performance in two critical ways: (1) by reducing model size, allowing for more simulations within a given computation time, and (2) by simplifying the learning problem, leading to faster learning and improved performance in fewer episodes. These findings highlight the potential of abstraction to address key challenges in BRL for F-POMDPs and open a promising direction for future research into leveraging abstractions to improve scalability, exploration, and decision-making in complex, real-world environments.

## 2 Background

In this section, we introduce the rich body of literature that our work builds upon. In particular, section 2.1 introduces the POMDP as the general mathematical model for sequential decision-making. We then describe (model-based) Bayesian reinforcement learning in factored POMDPs in section 2.2, which is a (belief-space) Partially-observable Markov decision process itself. Lastly, we discuss algorithmic approaches for solving these decision problems in section 2.3.

### 2.1 POMDPs and Factorization

Sequential decision-making in stochastic domains with hidden states can be formalized as a POMDP (Boutilier & Poole, 1996; Kaelbling et al., 1998). The POMDP is defined by the tuple $(\mathbb{S}, \mathbb{A}, \mathbb{O}, \mathcal{D}, \mathcal{R}, \gamma, H)$, where $\mathbb{S}$, $\mathbb{A}$, and $\mathbb{O}$ are the (discrete) set of states, actions, and observations, respectively. The dynamics $\mathcal{D}$ specify the system's transition probabilities $\mathcal{D} \in \mathbb{D}$: $(\mathbb{S} \times \mathbb{A}) \to \Delta(\mathbb{S} \times \mathbb{O})$, and $\mathcal{R}$: $(\mathbb{S} \times \mathbb{A} \times \mathbb{S}) \to \mathbb{R}$ is the reward function. The (maximum) number of time steps in an episode is the horizon $H \in \mathbb{Z}$, and $\gamma \in [0, 1]$ is the discount factor.

The goal of the agent is to maximize the discounted return, $\sum_t \gamma^t r_t$. To do so, it can use the observable action ($a \in \mathbb{A}$) and observation ($o \in \mathbb{O}$) history $h_t = (a_0, o_1, \ldots, a_{t-1}, o_t)$, or it can use the belief as a sufficient statistic. The belief is the probability distribution over the current state $b \in \mathbb{B}$: $\Delta\mathbb{S}$, which can be updated with Bayes' rule: $b'(s') = \tau(b, a, o)(s') \propto \sum_s \mathcal{D}(s', o|s, a)b(s)$. However, in most problems, the computation of the belief update is intractable. Section 2.3 will cover how to find a solution.

**Running Example** As a simple intuitive example, consider the Corridor domain shown in fig. 1a. The agent starts at the "Start" location and its goal is to reach the "Reward" location. As depicted in the figure, there is also a "Boots" location and a "Button" location. To reach the reward the "Door" must be open. There is also always a "Person" present in the environment. There are 8 different persons, with exactly one present during each episode. The probability of the person opening the door at some point during an episode varies depending on which person is present, but this probability is generally very low, ranging from about 1.25% to 10%. The agent cannot interact with the person.

The state consists of the location of the agent, the person present and the binary statuses of the boots, button, and door. The agent has four actions: move *left* or *right*, *put on boots*, *push button*, and *lock pick* the door. Moving left or right succeeds 30% of the time without boots and 95% of the time with boots. The probability of success for both *put on boots* and *push button* is 90%, provided the agent is in the correct location. When the button is pressed, there is a 100% chance that the door opens. The *lock pick* action only has an effect when the agent is standing next to the door and has a 40% chance of success. The agent can also open the door by "bashing" into it, specifically by using the move right action to collide with the door. This method is not very effective and has only a 5% chance of opening the door. The state is not fully observable to the agent; instead, it receives a noisy observation, which will be explained later.

**Factorization** The dynamics of POMDPs can often be captured efficiently through factorization and graphical models, such as Dynamic Bayes networks (DBNs) (Boutilier et al., 1999; Murphy, 2002). Throughout this work, the graphical model we rely on is specifically a two-stage DBN (Boutilier et al., 1999). Such

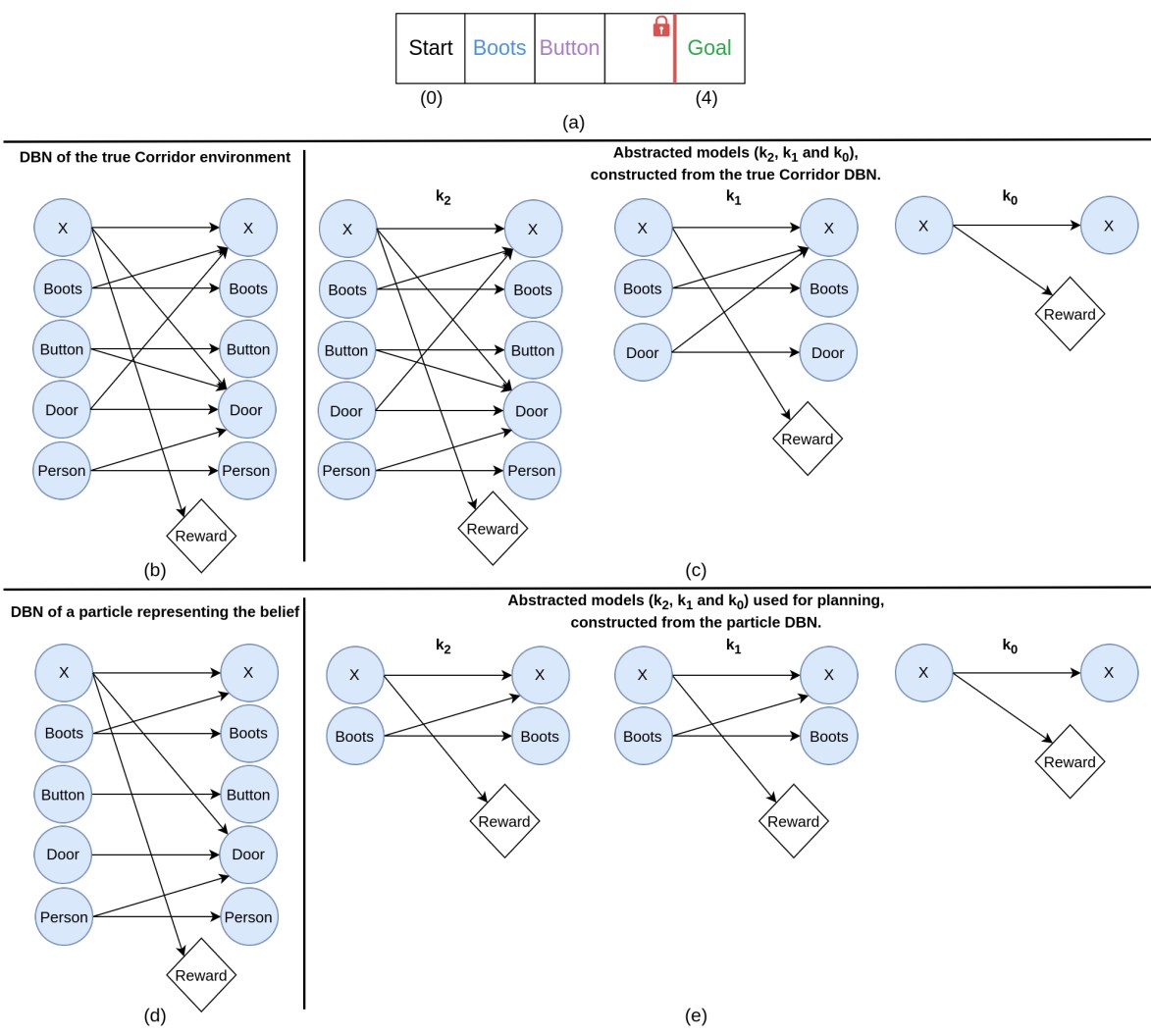

Figure 1: a) The Corridor domain. b) ground truth graph representing the dynamics of the Corridor domain for the action *right*. All the factors are partially observable, the agent gets an independent but noisy observation of each state factor. c) The abstract models for different levels of abstraction (denoted by $k_0, k_1, k_2$), constructed from the true Corridor DBN for the action *right*. d) We use particle filters to represent the belief, each particle contains a DBN of the corridor, this is an example of such a DBN, and e) the abstract models constructed from this example DBN.

models represent random variables by their features (also referred to as factors) and provide the ability to capture independence between these features. We assume that the full set of state factors and their (discrete) set of possible values is known.

A Bayes network (BN) is defined by a topology and conditional probability tables (CPTs). The topology $G \in \mathbb{G}$ describes the structure of the nodes in the graph, whether there is a dependency between pairs of nodes, where $\mathbb{G}$ is the set of all possible edge configurations. These directed edges define the *parents* $Pa(x_i'; G)$ of each node $x_i'$ as the set of incoming nodes (where we typically drop the dependency on the topology $G$ in the notation). The CPTs $\theta \in \Theta$ govern the conditional probability distribution of a node $x_i'$ given the full set of nodes $x : p(x_i'|x) = p(x_i'|Pa(x_i'; G); \theta_i)$.[1] In discrete environments, for example, these typically are categorical distributions; one for each parent value combination for each node. The *Dynamic* BN (DBN) restricts the space of graph topologies by allowing only directed edges from variables at one time

---

[1]Here, $x$ denotes the full set of nodes, and $Pa(x_i'; G) \subseteq x$ is the subset that directly influences $x_i'$.

step to variables at the next. This is convenient for (Markovian) dynamic systems, where random variables change over time. The Factored POMDP (F-POMDP) factorizes the state and observation space into nodes and describes the dynamics with a DBN for each action (Boutilier & Poole, 1996) (e.g., see fig. 1 as a DBN of the running example). In this work, we propose to learn abstract models that remove factors from the graph based on the reward node's dependency on them.

**Bayesian inference over DBNs** When the parameters of a model are not given, the Bayesian approach is to assume a prior (to compute posteriors) instead. The DBN is defined by — and thus a prior must describe a probability distribution over — its *topology* and CPTs. The prior over the topology assigns a (prior) probability to each graph structure: the probability that a next state and observation factor depends on a current state factor. The CPTs are categorical distributions and, hence, the Dirichlet distribution is a natural prior (Frigyik et al., 2010). Dirichlet distributions are parameterized by a collection of conditional count tables (CCCT), with one conditional count table (CCT) $\chi \in X$ for each unique set of parent values for each node.

Given initial counts specified by a prior and data from a categorical distribution, the posterior is again a Dirichlet (with new counts CCCT). In particular, given a topology $G$, prior CCCT and a new data point $(x, x')$, the Bayesian posterior is computed by incrementing the count $\chi_{x', x[Pa(x'_i)]}$ of each node $x'_i$ that is associated with its parent's values $x[Pa(x'_i)]$. This incrementing operation on POMDP transitions will be used frequently and we denote updating counts $\chi \in X$ given a transition $(s, a, s', o)$ with $\mathcal{U}$: $(X \times \mathbb{S} \times \mathbb{A} \times \mathbb{S} \times \mathbb{O}) \to X$. Note that there is no closed-form solution to the posterior over topologies.

## 2.2 Factored BA-POMDPs

*If* the state transitions were *not* hidden, one could simply maintain a set of the counts CCTs associated with each transition and over time converge to the true dynamics. This is the case under full observability (MDPs), and is called the Bayes-Adaptive MDP (BA-MDP) (Duff, 2002). Unfortunately, this is not the case in partially observable environments[2], and hence, there is uncertainty over these counts $\chi$.

The Factored Bayes-Adaptive POMDP (FBA-POMDP) (Katt et al., 2019) captures this uncertainty by using the POMDP formalism. In particular, this Bayes-adaptive model *is* a POMDP whose state space consists of both the state and the dynamics of the original POMDP (Ross et al., 2011). Formally, the FBA-POMDP is a tuple $(\dot{\mathbb{S}}, \mathbb{A}, \mathbb{O}, \dot{\mathcal{D}}, \dot{\mathcal{R}}, \gamma, H)$, where $\dot{\mathbb{S}}$ is the augmented state space: $\dot{\mathbb{S}} = \mathbb{S} \times \mathbb{G} \times X$. I.e., each (hyper-)state $\dot{s} \in \dot{\mathbb{S}}$ contains a domain state $s$, a topology $G$, and a CCCT $\chi$: $\dot{s} = \langle s, G, \chi \rangle$.[3] The action and observation spaces, the horizon, and the discount factor are taken directly from the original POMDP. Similarly, the reward function relies on the underlying system: $\dot{\mathcal{R}}(\dot{s}, a, \dot{s}') = \mathcal{R}(s, a, s')$. Note that, while this function is typically assumed known in BRL, we also conduct experiments in which this too is uncertain. Lastly, the dynamics $\dot{\mathcal{D}}$ dictate how augmented states transition:

$$\dot{\mathcal{D}} = p(s', o, G', \chi'|s, a, G, \chi) \tag{1}$$

$$= p(s', o|s, a; G, \chi)\mathbb{1}_G(G')\mathbb{1}_{\chi'}(\mathcal{U}(\chi, s, a, s', o)), \tag{2}$$

where $p(s', o|s, a; G, \chi)$ can be written as:

$$p(s', o|s, a; G, \chi) = p(s'|s, a; G, \chi)p(o|a, s'; G, \chi). \tag{3}$$

In equation 2 the term $p(s', o|s, a; G, \chi)$ shows that the model $(G, \chi)$ in state $\dot{s}$ determines the probabilities of the next state $s'$ and observation $o$. The $\mathbb{1}.(\cdot)$ is the indicator function, and encodes that there is only one non-zero transition, namely where the next topology equals the previous $G' = G$ and the next counts are increments of the previous according to $\mathcal{U}$. Since the POMDP is fully specified, the original learning problem is cast to a planning problem *with known* dynamics, given a prior $p_\mathcal{D}$. Most importantly, the exact solution

---

[2]In an MDP we see the whole transition (s,a,s'), so we can update the count $\chi(s, a, s')$. In contrast, both $s$ and $s'$ are hidden in the partially observable case. So we have to update $\chi(s, a, s')$ based on our belief resulting from the action-observation history rather than the real transitions.

[3]In Section 3, we extend these hyperstates to incorporate an abstract model, comprising an abstract topology $\bar{G}$ and a corresponding abstract set of counts $\bar{\chi}$.

to this planning problem yields the optimal policy, in terms of exploration-exploitation, with respect to the prior (Ross et al., 2011). Now we can apply our standard POMDP planning tools (e.g., particle filtering, planning) to FBA-POMDPs, as anytime solvers provide good approximations which converge to the exact solution in the limit of infinite compute (Silver & Veness, 2010; Katt et al., 2017; 2019).

## 2.3 Solving FBA-POMDPs

Unfortunately, FBA-POMDPs are very large, and naive applications of planning techniques will fail. Specifically, methods that require exact belief updates cannot be directly applied, as these updates are only feasible with a finite POMDP representation, which is impractical for large problems. This limitation makes it difficult to apply traditional POMDP planning methods without significant modifications. We give a high-level description of how solutions for FBA-POMDPs can be found, and refer to the original work (Katt et al., 2017; 2019) for details. Just like in any other POMDP, a planning solution requires two components: belief tracking and action selection.[4]

**Belief tracking in the FBA-POMDP**  The belief, the posterior over the current state, is a probability distribution over the POMDP state and its distribution $b \in \Delta(\mathbb{S} \times \mathbb{D})$ given the observed history $h_t = (a_0, o_1, \ldots, a_{t-1}, o_t)$. Unfortunately, the computation of the belief update is intractable in most problems. Thus, the belief is often approximated with particles instead (Thrun, 1999). A particle filter represents a distribution through *particles*, which in this case represent FBA-POMDP states, specifically each particle is a weighted FBA-POMDP state $(w, s, G, \chi)$ with (unnormalized) weight $w \in \mathbb{R}^+$. There are numerous sampling mechanisms for updating the belief given a new action-observation pair (Thrun, 1999), but in this work we applied sequential importance sampling and re-invigorate the belief with a Metropolis-Hastings-within-Gibbs sampling procedure (Katt et al., 2019) when necessary (details in appendix A). The initial particles are sampled from a prior belief, which may encode varying degrees of domain knowledge. Across experiments, we consider different settings in which parts of the DBN structure or certain CPTs are assumed to be known, while others are learned from data. Details on the prior knowledge used in the experiments can be found in section 4 and appendix D.

**Action selection in the FBA-POMDP**  Even with approximated belief updates, the belief space can be very large, especially in high-dimensional problems. Thus it is often infeasible to compute the action that maximizes the discounted return for every possible belief the agent could end up in. As a result, we extend the planner for FBA-POMDP (Katt et al., 2019) to pick actions *online* instead. Like any Monte-Carlo tree search (MCTS) method, this method incrementally builds a look-ahead tree of simulated interactions in the (FBA-POMDP) environment. Each iterations samples a (hyper) state $(s, G, \chi) \sim b$ and simulates an interaction in the FBA-POMDP, where actions are picked according to Upper Confidence Bounds Applied to Trees (UCT) (Kocsis & Szepesvári, 2006) to trade-off exploration and exploitation. For more details, see (Browne et al., 2012) for a survey on MCTS, (Silver & Veness, 2010) for MCTS in POMDPs, and (Katt et al., 2017; 2019) for MCTS in Bayes-Adaptive POMDPs (BA-POMDPs).

## 2.4 State abstraction for (factored) MDPs

State abstraction can be used to simplify complex problems by mapping the original state space to a smaller abstract state space (Li et al., 2006). This mapping is defined by an abstraction function $\phi$, which maps each state $s$ to an abstract state $\bar{s}$, where the bar notation indicates the abstract state space. Related to our abstraction approach is the notion of model-similarity abstraction. This comes in both an exact form, model-irrelevance abstraction (Li et al., 2006), and an approximate form, approximate model-similarity abstraction (Abel et al., 2016). These abstractions are also known as (approximate) stochastic bisimulation (Dean et al., 1997; Givan et al., 2003).

In the exact case, states are grouped if and only if they yield identical rewards and transition functions in the abstract space under all actions. Formally, in a model-irrelevance abstraction, $\phi(s_1) = \phi(s_2)$ if and only

---

[4]The original POMCP (Silver & Veness, 2010) implementation combined these steps to some extent, but we separate them out.

if

$$\forall_{a \in A} \; R(s_1, a) = R(s_2, a), \tag{4}$$
$$\text{and } \forall_{\bar{s}' \in \bar{S}} \; T(\bar{s}'|s_1, a) = T(\bar{s}'|s_2, a), \tag{5}$$

where the transition probability to an abstract state $\bar{s}'$ is given by $T(\bar{s}'|s,a) := \sum_{s' \in \bar{s}'} T(s'|s,a)$. In the approximate case, these equalities are relaxed, requiring that the reward functions and transition probabilities to any abstract state $\bar{s}'$ differ by no more than a small parameter $\eta$. Exact abstractions preserve optimality, that is, a solution in the abstract MDP is an optimal solution in the original MDP. Approximate abstractions do not preserve optimality, but have a bounded loss in value based on the value of $\eta$ (Abel et al., 2016).

In Factored MDPs (F-MDPs), Dearden & Boutilier (1997) suggest leveraging the structure of the problem to remove factors which are less relevant or irrelevant. An abstract MDP can be constructed from the remaining factors. Such an abstraction can be viewed implicitly as a mapping function $\phi$, where each configuration of the retained factors defines an abstract state, and all corresponding combinations of the removed factors are mapped to this same abstract state.

## 3 Abstraction for FBA-POMCPs

FBA-POMCP is able to learn and exploit the structure in POMDPs in a Bayesian way. However, it struggles when the number of factors grows large. On the one hand, the presence of many state factors itself slows down sampling, potentially leading to insufficient simulations to derive adequate actions. On the other hand, the prior belief over models with many state factors typically have low probability for models in which all factors have a small number of parents. As a consequence, the particle filter typically contains models that have at least a few factors with many parents. This leads to slow learning (low statistical strength) and possibly to exploration of factors with little or no effect on the rewards. These issues are problematic because our primary focus is on the task performance, rather than on learning the correct model itself.

A natural idea, therefore, is to explore in how far abstraction can address these two issues. While abstractions could lead to inaccurate models, in the regular (non-Bayes adaptive) planning case, it has been demonstrated that abstracting away factors with a weak influence can still improve the performance of online planning (He et al., 2020). Further, without abstraction, the agent could waste time exploring the dynamics of factors with little impact on the performance, if they are falsely believed to be influential. By removing these factors, abstraction can reduce unnecessary exploration and focus on the factors relevant for performance. As such, we propose to explore the impact that abstraction of state factors can have when learning in partially observable settings, formalized as FBA-POMDPs.

Specifically, we propose to perform the Partially Observable Monte-Carlo Planning (POMCP) simulations with an abstracted FBA-POMDP model. We hypothesize that such abstraction can improve performance by 1) increasing the number of simulations that can be done thus improving performance in online planning, and 2) reducing unnecessary exploration of factors with little impact on the performance, and allowing to focus exploration on the relevant factors.

We cover the combination of abstraction with FBA-POMCP in four parts. First, we give a high-level overview of the abstraction method and how it is added to FBA-POMCP. Second, we define abstractions on different levels, denoted by $k_0, k_1, ...$, where $k_0$ represents the coarsest abstraction. We define these through *subsets of state factors* and show how to generate a subset of state factors from a graph structure for a particular level of abstraction. Third, we show how to use the subset of state factors to construct the abstract structure and counts. Finally, we provide theoretical support for the combination of FBA-POMCP with abstraction.

### 3.1 Adding Abstraction to FBA-POMCP

We propose a method to enable the FBA-POMDP framework to benefit from abstraction. Specifically, we use abstract models for online planning with a variant of FBA-POMCP. To operationalize this, we cover the following steps:

1. We expand the representations to include abstract states.

**Algorithm 1** Initialize Abstract Particle Filter

1: **Input:** $p_{\dot{s}_0}$: prior over initial state
   $n$: number of desired particles
   $k$: abstraction level
2: **for** $i \in 0, \ldots, n$ **do**
3:     $\langle s_i, G_i, \chi_i \rangle \sim p_{\dot{s}_0}$
4:     $\langle \bar{s}_i, \bar{G}_i, \bar{\chi}_i \rangle \leftarrow \text{Abstract}(k, \langle s_i, G_i, \chi_i \rangle)$
5:     $w_i \leftarrow \frac{1}{n}$
6: **end for**
7: **return** $\{\bar{s}_i, \bar{G}_i, \bar{\chi}_i, w_i\}_{i=0}^n$

**Algorithm 2** Abstract

1: **Input:** $k$: abstraction level
   $\dot{s} = \langle s, G, \chi \rangle$: hyper-state
2: $Q \leftarrow \text{GetSubsetK}(k, G)$
3: $\bar{G} \leftarrow G$
4: $\bar{\chi} \leftarrow \chi$
5: **for** $(q, a) \in Q \times A$ **do**
6:     $\bar{G}, \bar{\chi} \leftarrow q.\text{MarginalizeCounts}(\bar{G}, \bar{\chi}, Q, a)$
7: **end for**
8: **for** $x \in G - Q$ **do**
9:     $\bar{G}.\text{remove}(x)$           // Remove factor
10:    $\bar{\chi}.\text{remove}(x)$           // Remove factor
11: **end for**
12: **return** $\langle \dot{s}, \bar{G}, \bar{\chi} \rangle$

2. While we do not exploit abstraction in the belief update, the abstracted belief state still needs to be updated. We cover the necessary modifications to the belief update process.

3. Finally, we explain how FBA-POMCP can use abstracted states.

**Expanding the Belief Representation**    When initializing the weighted particle filter in FBA-POMCP, each particle is a hyper-state $\dot{s} = \langle s, G, \chi \rangle$ associated with a weight $w$. The hyper-state $\dot{s}$ contains a ground state $s$, a graph structure $G$, and a set of counts $\chi$. For the initialization we require a probability distribution over the possible starting states, over the possible structures, and a probability distribution over the counts given a structure. For the running example, fig. 1d shows a possible structure of a hyper-state, in this case the factor *Door* is not believed to influence the $x$ factor, and the factor *Button* is not believed to influence the *Door* factor. When combining FBA-POMCP with abstraction, we abstract $G$ and $\chi$ and add the resulting abstracted structure $\bar{G}$ and counts $\bar{\chi}$ to each hyper-state $\dot{s}$. This leads to an abstract hyper-state: $\bar{s} = \langle \dot{s}, \bar{G}, \bar{\chi} \rangle$. The particle filter thus stores both the original hyper-state $\dot{s}$ and the abstracted structure $\bar{G}$ and counts $\bar{\chi}$. The construction of the abstract hyper-states is done during the initialization of the particle filter, as shown in algorithm 1.

For our running example, fig. 1e illustrates the structure $\bar{G}$ for different levels of abstraction, corresponding to the original structure $G$ in fig. 1d. In algorithm 1, the function Abstract creates the abstract model from a hyper-state $\dot{s}$, as detailed in algorithm 2. In the following sections we elaborate on the methods for selecting a subset based on the level of abstraction $k$ and on creating the abstract $\bar{G}$ and $\bar{\chi}$.

**The Belief Update Process**    To ensure consistency of the belief, we use the full model ($G$ and $\chi$) during the belief update (Russell & Norvig, 2016; Katt et al., 2019), as described in sections 2.2 and 2.3. This means that the belief update process remains largely the same. The main difference is that we now track and update both the full and abstract models, as shown in algorithm 3. As in section 2.2, the graph structures of both the full and abstract models stay the same during the update. The counts $\chi$ are updated via $\mathcal{U}$. Since each state maps to exactly one abstract state, the abstract counts $\bar{\chi}$ can be updated through the update of the counts $\chi$. Essentially, the (abstract) graphs remain unchanged, while the (abstract) counts are updated.

**Using Abstracted States in FBA-POMCP**    The planning process exclusively uses abstract models. Specifically, in Algorithm 8, the abstract representation is used to perform the environment *Step* function, which is utilized during simulations and roll-outs in the look-ahead tree search. The *Step* function uses the hyper-state with the abstract model and an action to sample a next abstract state by iteratively sampling the factors. It is shown in Appendix B.

The benefit of using an abstract model is that it speeds up planning since it contains fewer state factors, allowing for fasting sampling of the next state. When there is limited time for planning, this is one way in

---

**Algorithm 3** SIS with Abstraction

1: **Input:** $\{\dot{s}, \bar{G}, \bar{\chi}, w\}_{i=0}^n$: current (weighted) filter
          $a, o$: action and observation
2: **for** $i \in 0, \ldots, n$ **do**
3:     $s_i' \sim p(\cdot | s_i, a; G_i, \chi_i)$
4:     $w_i' \leftarrow w_i \times p(o | s_i', a; G_i, \chi_i)$
5:     $\chi_i' \leftarrow \mathcal{U}(\chi_i, s_i, a, s_i', o)$
6:     $\bar{\chi}_i' \leftarrow \bar{\mathcal{U}}(\bar{\chi}_i, s_i, a, s_i', o)$
7: **end for**
8: // Normalize & re-sample
9: **return** $\{s_i', G_i, \chi_i', \bar{G}_i, \bar{\chi}_i' w_i'\}_{i=0}^n$

---

**Algorithm 4** GetSubsetK

1: **Input:** $k$: abstraction level
            $G$: Graph structure.
2: $Q \leftarrow$ GetMinimumSet()    // $k_0$, the IR factors
3: **if** $k == 0$ **then**
4:     **return** $Q$
5: **end if**
6: $Q' \leftarrow Q$
7: **for** L = 1; L $\leq$ k; L++ **do**
8:     **for** $(q, a) \in Q \times A$ **do**
9:        $Q' \leftarrow Q' \cup G.\text{getNode}(q, a).\text{parents}()$
10:     **end for**
11:     $Q \leftarrow Q'$
12: **end for**
13: **return** $Q$

---

which abstraction can improve performance. It is important to note that the abstract models are constructed at the beginning of the agent's lifetime, when its belief is initialized. As a result, these abstract models are always available.

## 3.2 Abstraction Via Subsets of State Factors

We introduce a method for performing abstraction in the Bayesian RL (BRL) context. Since we are interested in understanding how abstraction impacts the learning process, we base our approach on a relatively simple planning method for F-MDPs (Dearden & Boutilier, 1997), which is easy to understand and analyze. We make three important adaptations: 1) we make it applicable to partially observable problems, 2) we extend it to the RL setting, where the abstraction is not only used for planning but also for learning, and 3) we incorporate the counts required in BRL. Our approach introduces a level of abstraction that determines the factors to include based on the graph structure.

We define different levels of abstraction based on their connection to the reward in the graph structure. Each abstraction level is defined by a set of state factors that is included in the model, observation factors are always kept in the model. After abstraction, this can lead to observation factors without parents, we explain how we deal with this in section 3.3.4. We start building abstractions from the factors directly influencing the reward, the immediately relevant factors (IR). We first give a formal definition and then illustrate it with an example.

**Definition 1.** *The set of immediately relevant factors (IR) contains only the factors $q \in G$ that directly influence the reward. Specifically, these are the factors that are parents of the reward, denoted as $Pa(reward)$. The smallest subset of state factors $k_0$ is equal to IR, $k_0 = IR$. The set $k_i$ is the smallest set such that the following holds:*

     *1. $k_{i-1} \subseteq k_i$.*

     *2. If $q \in k_{i-1}$ then $Pa(q) \in k_i$.*

*The set $k_{inf}$ refers to the full model.*

In the running example, we see an example of a structure in fig. 1d. In this problem, the agent only receives a reward when it is in the goal location, i.e., $x = 4$. This is reflected in the graph structure where the only parent factor of the reward is $x$. In this case, $x$ is the only IR factor and the abstraction $k_0$ only contains $x$ as shown in fig. 1e. To construct the subset of state factors for $k_n$, we add the parents of the factors in $k_{n-1}$.

So to see which factors to include in $k_1$ in this example, we check in fig. 1d which factors are parents of $x$. In this case, that is only *Boots*. Finally, for $k_2$, no new factors are added since in the structure in fig. 1d the parents of *Boots* do not include any factors not yet in the set of $k_1$.

The procedure to get the subset of state factors for a given level of abstraction $k$ and a particle (or hyper-state) $\dot{s}$ is shown in algorithm 4. First, it initializes a set $Q$ with the set IR, retrieved with *GetMinimumSet*. For abstraction level $k_0$, this is what is returned immediately. For higher levels, it then builds the subset incrementally by adding the parents of the factors in Q. That is, to construct the subset of state factors $k_n$, it starts with $k_{n-1}$ and then adds the parents of these factors.

When the reward function is known, the function GetMinimumSubset directly returns the set IR. When the reward function is unknown, the reward itself is also modeled as a state factor that takes the same value as the reward. Uncertainty about the reward function can then be incorporated in the belief, and hyper-states may end up with different graph structures for the reward. The IR can then be retrieved by finding the parents of the reward state factor. We demonstrate that our method can deal with uncertainty about the reward and IR in section 4.4.

### 3.3   Abstract Model Construction

After retrieving a subset of state factors, we construct the abstract model. Since the abstraction uses only a subset of the state factors, this involves removing factors, and therefore we need to decide how to treat factors that have missing parents as a result of this. To illustrate, when we abstract a full model such as the one in fig. 1d to create the abstract model on level $k_0$ ( fig. 1e), we remove *Boots* and create a distribution for $x$ with only $x$ itself as the parent factor. The question then is how we can define a CCT that does not depend on 'Boots' from the original one that does.

For probability distributions in known models it is logical to resort to marginalization. However, in section 3.3.1 we show that the problem is more deeply rooted: marginalization leaves us with a term that is difficult to specify since it depends not only on the values of its parents but also on the policy, and it can change over time. As such there is no fundamentally right approach to do this form of abstraction. Instead, novel ideas and approximate approaches are needed. We explore two initial ideas for this form of abstraction in sections 3.3.2 and 3.3.3. In section 3.3.2 we make an assumption on the abstraction and show that we can simply aggregate the counts in that case. In section 3.3.3, we motivate using approximate abstraction and discuss potential issues that arise with the approximation.

### 3.3.1   Marginalization of Probability Distributions

Before considering the case of CCTs, we treat the case of probability distributions. For a probability distribution, given a factor $X$ with a set of parents $\text{Parents}(X)$, we can marginalize out a parent $Y$ or a set of parents. For ease of notation, we show the marginalization for one parent, multiple parents can be removed by repeating this process:

$$P(x_i|\mathbf{z}) = \sum_{y_j} P(x_i, y_j|\mathbf{z}) \tag{6}$$

$$= \sum_{y_j} P(x_i \mid \mathbf{z}, y_j) P(y_j \mid \mathbf{z}), \tag{7}$$

where $\mathbf{z} = (z_1, z_2, \ldots, z_n)$ denotes the realization of $\text{Parents}(X) \setminus Y$, while $x_i$ and $y_j$ represent a specific realization of the factors $X$ and $Y$. The term $P(y_j \mid \mathbf{z})$ acts as a weight for the contribution of $P(x_i|\mathbf{z}, y_j)$ to the marginal distribution $P(x_i \mid \mathbf{z})$.

However, estimating $P(y_j \mid \mathbf{z})$ for a DBN is nontrivial. In the running example, consider the probability of moving from $x = 1$ to $x' = 2$ after taking the action *right*. The sampled model in fig. 1d only has the factors $x$ and *Boots* as parents of $x$, and the abstract model $k_0$ (fig. 1e) does not include *Boots*. Following equation 7,

we can obtain the marginal distribution for $x$ by summing over the separate values of *Boots*:

$$P^{right}(x'|x) = \sum_{b \in Boots} P^{right}(x'|x, b)P(b|x). \tag{8}$$

However, while the probabilities $P^{right}(x'|x, b)$ are well defined (e.g., $P^{right}(x' = 2|1, b = \text{On}) = 0.95$), the value of $P(b|x)$ is not as clear. This probability represents the likelihood that the boots are on given a specific location $x$. However, it does not depend solely on $x$ itself. For instance, we might know that the probability of the boots being on is 0 at the start of the episode, but this probability generally depends on the history of actions and observations. In general, we can make the following observation:

**Observation 1.** *Accurately estimating $P(y|\mathbf{z})$ without additional information is generally not possible. This is because $y$ can depend on other variables, including itself, and on the policy that can change over time.*

The view of $P(y|\mathbf{z})$ as a weight in equation 7 is related to the concept of a weighting function in work on state abstraction (Dearden & Boutilier, 1997; Li et al., 2006). Theoretical work shows that, for some abstractions, a policy based on the abstract model (with any weighting function) can perform well in the real problem in planning (Li et al., 2006; Abel et al., 2016; Congeduti & Oliehoek, 2022) and in RL (Starre et al., 2023). For probability distributions, Dearden & Boutilier (1997) use a sort of average of the probabilities but also remark this can lead to suboptimal solutions. The best way to approach estimating $P(y|\mathbf{z})$ for the optimal performance is still an open problem.

### 3.3.2 Beliefs and Aggregating Counts for Exact Abstractions

In the previous section we discussed the problem of dealing with conditional probability tables (CPTs) where parents are abstracted, leading to dependence on the policy and history for estimating $P(y|\mathbf{z})$ in equation 7. In this section, we begin by considering a simplified setting where the abstraction is assumed to be exact. This assumption allows us to sidestep the difficulty of estimating $P(y|\mathbf{z})$ and to introduce the CCCT more cleanly under idealized conditions. This assumption is made only in this subsection to clarify the conceptual difference between exact and inexact abstractions. The more realistic and interesting case is when the assumption does not hold, which we address in section 3.3.3.

One situation where abstraction makes sense is when the model is overspecified; it contains links that are unnecessary. This is the case when the abstract model probabilistically behaves in the same way as the full model, which is what we assume in this subsection:

**Assumption 1.** *The abstraction is exact. That is, let $Z$ be the set of removed parents for a factor $X$, let $\mathbf{z} = (z_1, z_2, \ldots, z_n)$ denote a specific realization of the remaining parents $Parents(X) \setminus Z$, and let $\mathbf{z}_1^{removed}$ and $\mathbf{z}_2^{removed}$ be two different realizations of the removed parents $Z$. Then, for all realizations $x_i$ of $X$, we assume:*

$$P(x_i|\mathbf{z}) = P(x_i|\mathbf{z}, \mathbf{z}_1^{removed}) \tag{9}$$

$$= P(x_i|\mathbf{z}, \mathbf{z}_2^{removed}). \tag{10}$$

This assumption implies that the links between the removed parents and the corresponding child node were obsolete. For instance, consider a change in the running example where the boots would have no effect on $x$, then this would mathematically mean that $P(x'|x, \text{Boots} = \text{On}) = P(x'|x, \text{Boots} = \text{Off})$. We observe:

**Observation 2.** *Under assumption 1, $P(y_j|\mathbf{z})$ has no influence. That is, since*

$$P(x_i|\mathbf{z}) = \sum_{y_j} P(x_i|\mathbf{z}, y_j)P(y_j|\mathbf{z}) \qquad (equation\ 7) \tag{11}$$

$$= \forall_{y \in Y} : P(x_i|\mathbf{z}, y). \tag{12}$$

Thus, if boots had no influence, data collected with boots on and off can be used to estimate $P(x'|x)$. For example, consider the conditional counts $\chi(x'|x = 1, \text{Boots})$ in table 1.

Observation 2 means that, to marginalize in the CCCT, we no longer have to be concerned about $P(y_j|\mathbf{z})$. Which means that in table 1 we can aggregate the counts in the columns, formally:

$$\bar{\chi}(x_i|\mathbf{z}) = \sum_{y_j} \chi(x_i|\mathbf{z}, y_j). \tag{13}$$

For example, to determine the conditional counts in table 1, we apply equation 13 to construct the abstracted

Table 1: Initial conditional count table, for $x = 1$ and action *right*.

| Boots | $x' = 1$ | $x' = 2$ |
|---|---|---|
| On | 2 | 8 |
| Off | 6 | 4 |

Table 2: Count table after aggregation, for $x = 1$ and action *right*.

| | $x' = 1$ | $x' = 2$ |
|---|---|---|
| | 2+6 = 8 | 8+4 = 12 |

Table 3: Count table after aggregation and normalization, for $x = 1$ and action *right*.

| | $x' = 1$ | $x' = 2$ |
|---|---|---|
| | $1/2 * 8 = 4$ | $1/2 * 12 = 6$ |

(or marginalized) counts $\bar{\chi}$ from the original counts $\chi$. This is done for every action by aggregating the counts as follows:

$$\bar{\chi}^{right}(x'|x) = \sum_{b \in Boots} \chi^{right}(x'|x, b). \tag{14}$$

Writing out equation 14 we obtain the counts $\bar{\chi}^{right}(x'|x = 1)$ after aggregation:

$$\bar{\chi}^{right}(x' = 1|x = 1) = \sum_{b \in Boots} \chi^{right}(x' = 1|x = 1, b) = 2 + 6 = 8, \tag{15}$$

$$\text{and } \bar{\chi}^{right}(x' = 2|x = 1) = \sum_{b \in Boots} \chi^{right}(x' = 2|x = 1, b) = 8 + 4 = 12. \tag{16}$$

The resulting conditional counts are shown in table 2. Note that now the resulting row has counts ($\{8, 12\}$) which are higher than the individual previous rows for Boots *On* ($\{2, 8\}$) and *Off* ($\{6, 4\}$). This implies that after abstraction we are (relatively) more confident about these transitions than before. Under assumption 1 this does not have a large influence when the prior is close to the true distribution, as in that case these estimates should be close together. However, this could be different when the abstraction is not exact or if the prior is not close to the true distribution.

### 3.3.3 Aggregating Counts for Approximate Abstractions

Previously, we assumed that the abstraction was exact. Of course, this may not always be the case, or it may not be necessary to make this assumption. There exist scenarios where one could argue for the use of *approximate* abstractions (Dearden & Boutilier, 1997; Abel et al., 2016; Starre et al., 2023). For example, in cases where a parent only has a small influence, abstracting these parents away can lead to faster learning (Starre et al., 2023). Additionally, reducing the size of the model through abstraction can enhance performance by facilitating faster planning (He et al., 2020).

However, when the abstraction is not exact, observation 2 no longer holds. Specifically, with an approximate abstraction, $P(x|\mathbf{z}, y)$ generally varies for different instantiations of $y$. Consequently, $P(y|\mathbf{z})$ does influence the result. In this case, abstracting a candidate model could result in a probability distribution that deviates significantly from the behavior of the candidate model.

As an example, consider again the running example where with *Boots = On* we have a probability of moving of 95% and with *Boots = Off* only 30%. Table 1 shows are initial estimates where *Boots* is still included, counts of $\{2, 8\}$ and $\{6, 4\}$ for *Boots = On* and *Boots = Off*, respectively. These are reasonably accurate with, if we translate the counts to probabilities, an expected 80% and 40% chance of moving, respectively.

However, when *Boots* is removed we see in table 2 this leads to counts of $\{8, 12\}$, or an expected probability of moving of 60%. With *Boots* being removed from the model the agent is highly likely to be in a state with *Boots = Off*, and thus this leads to an overestimation of the probability of moving for the agent.

As alluded to in the previous section, the abstraction also increases the confidence in the resulting counts. When the abstraction is not exact, we could say that the increased confidence in the resulting counts is not warranted, there is overconfidence. This overconfidence can slow down learning since it will take more experience to change the belief. For example, the change in table 1 of adding an extra observation to the row with *Boots = Off* has a relatively larger effect than adding one extra observation after aggregation in table 2.

This means that the proposed aggregation in equation 13 does not work as well when assumption 1 does not hold, since it can lead to incorrect estimations with a higher confidence. As such, we want to adapt the aggregation method. **It is still an open question what the best way to aggregate when using approximate abstraction.**

We propose a way to reduce the overconfidence in the resulting dynamics through a normalization scheme, which should lead to quicker learning in cases where the prior and abstraction are biased. Let $Z = Y_1, Y_2, \ldots, Y_n$ denote the set of removed parents. To normalize, we multiply each entry by

$$\frac{1}{\prod_{Y \in Z} |\text{dom}(Y)|}, \tag{17}$$

where $|\text{dom}(Y)|$ represents the number of values that the parent $Y$ can take. For example, in the case of the position $x$ from the running example, we have $|\text{dom}(x)| = 5$.

Using $\tilde{\chi}$ to represent normalized counts, applying the normalization factor equation 17 to equation 13 results in:

$$\tilde{\chi}(x_i|\mathbf{z}) = \frac{1}{\prod_{Y \in Z} |\text{dom}(Y)|} \sum_{(y_1, y_2, \ldots, y_n) \in Y_1 \times Y_2 \times \cdots \times Y_n} \chi(x_i|\mathbf{z}, y_1, y_2, \ldots, y_n). \tag{18}$$

Intuitively, the proposed normalization scheme reduces the counts proportionally to the amount of rows that is removed during aggregation. Applying equation 18 to the example where we remove *Boots* this leads to:

$$\tilde{\chi}^{right}(x' = 1|x = 1) = \frac{1}{|\text{dom}(Boots)|} \sum_{b \in Boots} \chi^{right}(x' = 1|x = 1, b) = \frac{1}{2}(2 + 6) = 4, \tag{19}$$

$$\text{and } \tilde{\chi}^{right}(x' = 2|x = 1) = \frac{1}{|\text{dom}(Boots)|} \sum_{b \in Boots} \chi^{right}(x' = 2|x = 1, b) = \frac{1}{2}(8 + 4) = 6, \tag{20}$$

also shown in table 3.

By applying the normalization of equation 17 the amount of counts in the table after aggregation is equal to the average amount of counts in the initial prior. For example, in table 1 the counts in the rows both sum up to 10, and the counts after the aggregation and normalization also sum up to 10 (table 3).

The proposed normalization provides a robustness against mistakes in the prior and approximate abstraction, by lowering the impact that the prior has on the learning. In our experiments, we investigate this normalization scheme, showing that this can significantly speed up learning.

### 3.3.4 The Observation Space for Abstract Models

Since we remove state factors from the model in the abstraction, a natural question is how we deal with the observation factors. We can consider two cases, 1) where a part of the parents is removed, and 2) when all the parents are removed.

In the first case, we simply perform the aggregation in the same way as for the state factors. It is the second state that poses a problem, as it leaves us with no parents for the observation factor. Since in this case the observation would provide no actual information about the underlying state, we enhance the observation space of the abstract model by including an observation "not observed" for all observation factors. For observation factors where all parents are removed the observation function simply returns "not observed".

### 3.4 Theoretical Support

Here we will show how the combination of FBA-POMCP with the abstraction method can lead to near-optimal performance when the abstraction is good. We first show that transforming the original FBA-POMDP with the abstraction method results in another FBA-POMDP. Because of this, the theoretical guarantees of FBA-POMCP apply to the abstracted problem. Then we give a definition for the quality of the abstraction that gives a guarantee on the performance in the original FBA-POMDP. Together this shows that abstractions leads to near-optimal solutions with FBA-POMCP, when a good abstraction is used.

First, we note that the abstraction results in another FBA-POMDP:

**Lemma 1.** *The result of applying the abstraction method, as described in algorithm 2 and section 3.3, to the original FBA-POMDP results in another FBA-POMDP, the abstract FBA-POMDP.*

In appendix C, we present a constructive proof. In essence, abstraction reduces the state space and marginalization produces a dynamics function for the resulting state space. Lemma 1 implies that we can use POMCP to find a near-optimal solution with respect to the belief in the abstract FBA-POMDP, due to the following result:

**Theorem 1** (Katt et al., 2019 ). *Given a belief $b(s, G, \chi)$, FBA-POMCP converges to an $\epsilon$-optimal value function of a FBA-POMDP: $V(b, a) \xrightarrow{p} V^*(b, a) - \epsilon$.*

The bias of the value function, $\epsilon$, can be made arbitrarily small by increasing the maximum search depth. If there is no limit on the search depth, the bias $\epsilon$ is $O(\frac{\log(n)}{n})$ in the limit of the number of simulations $n$ starting from the belief $b$ (Silver & Veness, 2010).

The question that remains is, how good is the solution for the abstract FBA-POMDP in the original FBA-POMDP? In general, the quality of a solution when using abstraction depends on the type and quality of the abstraction (Dearden & Boutilier, 1997; Abel et al., 2016). We define the quality $\eta$ of the abstraction as the upper bound of the difference between optimal value function of the original FBA-POMDP and the value function of the original FBA-POMDP under a $\epsilon$-optimal policy for the abstracted FBA-POMDP:

**Definition 2.** *An abstraction has a quality $\eta$, s.t. every $\epsilon$-optimal solution $\pi$ of the abstract FBA-POMDP applied to the original FBA-POMDP has suboptimality bounded by $\epsilon + \eta$:*

$$\forall (b, a) \in \mathbb{B} \times \mathbb{A} : \ |V^*(b, a) - V^\pi(b, a)| \leq \epsilon + \eta. \tag{21}$$

This definition is based on existing suboptimality bounds for abstract models such as the approximate model-similarity abstraction (Dearden & Boutilier, 1997; Abel et al., 2016), discussed in section 2.4. Informative bounds can be derived, for instance, by assessing how well the abstract model can approximate the original model (Dearden & Boutilier, 1997; Starre et al., 2023). In the case of approximate model-similarity abstraction, the parameter $\eta$ is small when the grouped states have similar transition and reward functions. Recent work shows that, in the context of learning with abstraction in MDPs, this type of abstraction can yield such bounds (Starre et al., 2023). Analogously, in POMDPs, similar results could be expected to hold if the dynamics of the abstract model closely resemble those of the true model. Extending the findings in the fully observable to the partially observable setting could be feasible by incorporating learning of the observation function and applying a simulation lemma for POMDPs (Lee et al., 2023).

When combined with Theorem 1, definition 2 implies that combining FBA-POMCP with abstraction leads to a near-optimal solution for the original FBA-POMDP, particularly when the abstraction effectively captures the dynamics of the original problem (i.e., when $\eta$ is small):

**Corollary 1.** *Given a belief $b(s, G, \chi)$ and an abstraction, assuming there exists an $\eta$ for which equation 21 holds, FBA-POMCP combined with abstraction converges to an $\epsilon + \eta$-optimal value function of the original FBA-POMDP: $V^\pi(b, a) \xrightarrow{p} V^*(b, a) - (\epsilon + \eta)$.*

*Proof.* By lemma 1 the problem after abstraction is still an FBA-POMDP. By theorem 1, we can use FBA-POMCP to get a policy $\pi$ within $\epsilon$ of the optimal solution of this abstract FBA-POMDP. Then, by definition 2 this leads to a solution within $\epsilon + \eta$ of the optimal solution of the original FBA-POMDP. $\square$

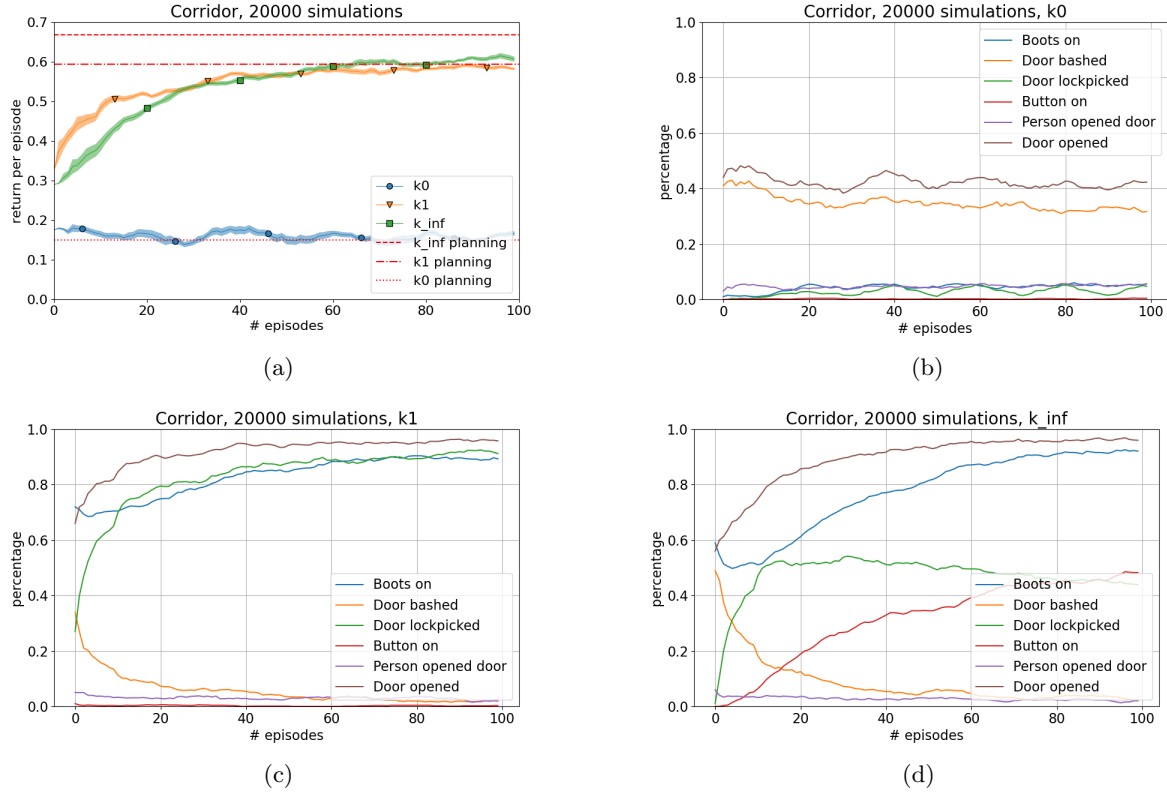

Figure 2: a) Performance in the Corridor domain. The learning behavior in the Corridor domain is shown for different models: b) $k_0$, c) $k_1$, and d) $k_{\inf}$. The y-axis shows the percentage of runs in which a certain behavior or occurrence happened during each episode.

This result shows that we can reach near-optimal performance in the original problem with good abstractions, but that better performance can be achieved in theory without abstraction when $\eta > 0$. However, in practice abstraction could perform better through faster simulations, and it could get to a good performance more quickly through greater statistical strength due to aggregation.

# 4 Experiments

We aim to investigate three questions; 1) does abstraction lead to faster simulations and enable scaling to more complex problems, 2) does abstraction lead to faster learning by reducing unnecessary exploration, and 3) does abstraction provide these benefits when the reward function is not known? We empirically evaluated our approach on four domains to answer these questions. The first domain is the Corridor problem from the running example, a simple problem where the trade-off of using abstraction is shown. The second domain is Cracky Pavement Gridworld where we show the advantage of abstraction in a very large problem, with a state space of up to size $|\mathbb{S}| = 6 \times 10^{25}$. In addition, in this domain we show the effectiveness of the proposed normalization step for abstraction, equation 17, in the $k_0$ model. The third domain is an adjusted version of Collision Avoidance (Katt et al., 2019; Luo et al., 2019), made more complex to allow for more abstraction. The final domain is Room Configuration, where the agent must learn the reward function. For further details on the experimental setup and the domains we refer to appendix D.

## 4.1 Corridor

The Corridor domain is the domain described in the running example in section 2.1, where there are 8 different *Persons* that can be present, each with a small probability of opening the door. The agent is

uncertain about the transition dynamics of the $x$-position, for the actions left and right, and the door. We evaluate the abstractions $k_0$, $k_1$, and the full model $k_{\inf}$. The abstraction $k_0$ only includes the factor $x$. Abstraction $k_1$ includes $x$ and, depending on the structures included in the belief, can also include the factors *Boots*, *Button* and *Door*.

**Results**  We test the effectiveness of two levels of abstraction under a fixed number of simulations. Figure 2a shows the simple moving average of the return per episode, and the shaded areas show the 95% confidence interval. Figures 2b to 2d illustrate the behavior of the different models across episodes. Specifically, these figures display how often the agent put on the boots, how often the door was opened, and through which means the door was opened. For example, in fig. 2d, it can be seen that in the first episode, the agent rarely lock picked the door (close to 0%), whereas after 20 episodes, the agent lock picked the door in more than 50% of the runs.

First, there is a large difference in the performances between $k_0$ and the other two models. This difference occurs because the full model and model $k_1$ can learn to open the door through more effective means than simply bashing against it, while $k_0$ cannot. This is due to the fact that the model $k_0$ only keeps the $x$ factor and does not include *Door*, *Boots*, and *Button*. Consequently, it does not recognize that the actions to open the door, push the button, and put on the boots have any effect. In addition, since these actions have no direct effect on the $x$-position, $k_0$ is unable to learn interactions with the environment beyond moving left and right. It ultimately learns a strategy of just moving to the right. This strategy can still lead to reaching the goal state, as there is a chance that the door opens when the agent bashes into it by moving right, or when the person opens the door. This can be observed in fig. 2b, which shows that most of the time the agent successfully opened the door by bashing into it. While this shows the agent never learns to open the door by pushing the button, it does occasionally open the door through the *lock pick* action. This can happen near the end of the episode, with just one step remaining when the agent is at location $x = 3$ and the door is still closed. In such situations, the values for the different actions in the tree search will all be zero. Since the action is then randomly chosen among those with the same value, this can lead to selecting the *lock pick* action.

The $k_1$ model can effectively learn to interact with a part of the environment because it keeps not only the $x$ factor but also the factors that influence $x$. This means that if the believed model is the correct model, the $k_1$ model also contains *Boots* and *Door*. Consequently, $k_1$ learns to *put on boots* and to *lock pick* the door, leading to a much better performance than $k_0$. Due to greater statistical strength from aggregation, $k_1$ learns to use lock pick more quickly than the full model.

Technically, it would be possible for $k_1$ to also open the door by *pushing the button* if the *Button* factor is also believed to influence $x$. However, as shown in fig. 2c, this did not occur frequently in the experiments. Instead, $k_1$ rapidly learns to *put on boots* and *lock pick*.

When comparing the performance of $k_1$ with the full model $k_{\inf}$, we can distinguish three phases. Initially, $k_1$ learns more quickly than $k_{\inf}$ because the full model $k_{\inf}$ takes longer to learn to *lock pick*. Then, $k_{\inf}$ catches up as it learns to *lock pick*. Finally, $k_{\inf}$ starts to surpass $k_1$ by learning to open the door through pushing the button, as is visible in fig. 2d.

This experiment shows that when the abstraction is not exact, it can initially still lead to better performance because of greater statistical strength due to aggregation. Since in this domain the abstract models cannot learn the optimal behavior, they eventually get outperformed by the full model.

## 4.2 Cracky Pavement Gridworld

The Cracky Pavement Gridworld is a grid world as shown in fig. 3a. The DBN of the domain is shown in fig. 3b. The state space is factored into the $x$ and $y$ locations, *Rain*, and 20 or 80 extra binary factors. Only $x$, $y$, and *Rain* influence the movement of the agent, though the agent must infer this. Movement success depends on tile type and rain conditions: movement is difficult on Trap tiles, and on the Vendor tile when it is dry, due to the presence of a vendor. When it rains, the vendor leaves and the tile becomes easier to traverse. The agent observes $x$ and $y$ noisily and does not know which factors affect transitions. We evaluate the abstractions $k_0$, $k_1$, and the full model $k_{\inf}$. The abstraction $k_0$ includes only the factors $x$

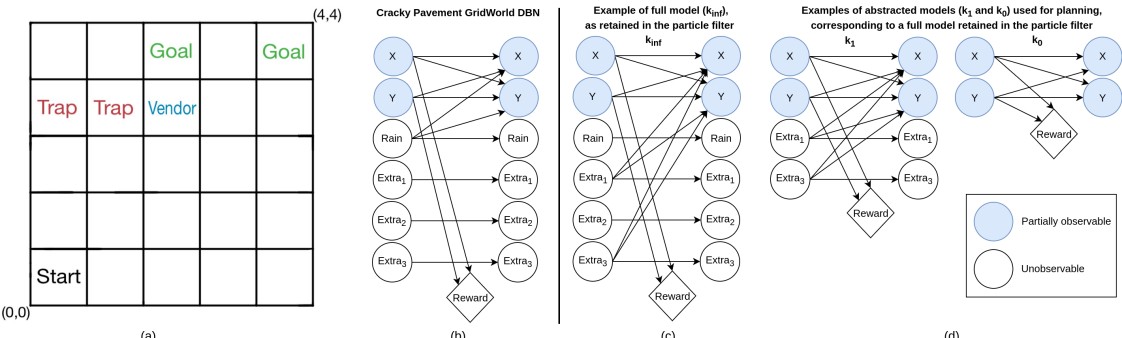

Figure 3: a) The Cracky Pavement Gridworld, b) ground truth graph representing the dynamics of the Cracky Pavement Gridworld problem with 3 extra binary factors, c) example of a full model in the particle filter, d) examples of two abstract models.

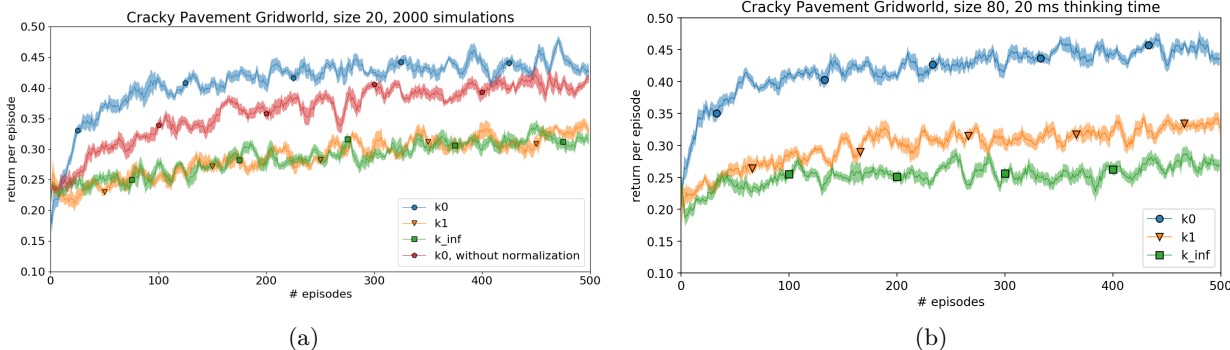

Figure 4: Performance in the Cracky Pavement Gridworld domain, a) with a fixed thinking time, b) with a fixed number of simulations.

and $y$ since they are the ones that directly influence the reward (fig. 3b). Abstraction $k_1$ also includes the parents of $x$ and $y$.

**Results** First, we examine the results for one particular setting, with 80 extra binary factors and 2000 simulations. Figure 4a shows these results, where the lines represent the simple moving average of the return per episode, and the shaded regions indicate the 95% confidence interval. Both $k_1$ and the full model $k_{inf}$ struggle to learn a good policy, while $k_0$ outperforms both.

The primary difference between $k_0$ and the other two models is that $k_0$ retains only the $x$ and $y$ factor, while $k_1$ includes any factor it believes influences $x$ or $y$, such as *Rain* or parts of the additional binary factors, and $k_{inf}$ retains all factors as it does not abstract. Although $k_0$ sacrifices the ability to account for *Rain*, it performs

Table 4: Average number of simulations in the Cracky Pavement Gridworld.

|  | Size 20 | | | Size 80 | | |
|---|---|---|---|---|---|---|
| Time | $k_0$ | $k_1$ | $k_{inf}$ | $k_0$ | $k_1$ | $k_{inf}$ |
| 5ms | 634 | 550 | 272 | 521 | 429 | 149 |
| 10ms | 1324 | 1263 | 573 | 1097 | 871 | 240 |
| 15ms | 1978 | 1658 | 660 | 2044 | 1726 | 328 |
| 20ms | 2571 | 2145 | 803 | 2433 | 2360 | 440 |

Table 5: Average return over the first 500 episodes in the Cracky Pavement Gridworld.

|  | Size 20 | | | Size 80 | | |
|---|---|---|---|---|---|---|
| Time | $k_0$ | $k_1$ | $k_{inf}$ | $k_0$ | $k_1$ | $k_{inf}$ |
| 5ms | 0.33 | 0.26 | 0.23 | 0.31 | 0.25 | 0.18 |
| 10ms | 0.39 | 0.29 | 0.26 | 0.37 | 0.28 | 0.21 |
| 15ms | 0.41 | 0.31 | 0.27 | 0.41 | 0.30 | 0.23 |
| 20ms | 0.42 | 0.30 | 0.28 | 0.42 | 0.30 | 0.25 |

better because it simplifies learning about the trap states by only considering $x$ and $y$. This leads to greater statistical strength as it is much easier to learn $P(x'|x, y)$ than $P(x'|x, y, \text{rain, and numerous binary factors})$.

While $k_0$ cannot distinguish between rain and no rain and therefore does not learn when the vendor is on the tile, it can learn to navigate around this tile. Although this is generally not optimal, it is optimal when rain and the vendor are not considered. The full model $k_{\text{inf}}$ and $k_1$ can eventually outperform $k_0$ by learning that it is better to cross the vendor tile when it is raining, as shown in appendix E. However, this takes a considerable amount of time, and during the earlier episodes, $k_0$ performs much better earlier.

These results shows that the greater statistical strength obtained by removing information, like *Rain* and extra binary factors, can result in a significant increase in performance. Additionally, fig. 4a compares the performance of $k_0$ with and without the proposed normalization step for abstraction (equation 18), demonstrating that the normalization step can significantly improve learning performance.

In fig. 4b, where we compare the performance with a fixed amount of thinking time instead of a fixed number of simulations, we see that $k_1$ outperforms the full model $k_{\text{inf}}$. The main difference between $k_1$ and $k_{\text{inf}}$ is that $k_1$ abstracts away all the factors that, given the graph topology, are not directly or indirectly relevant for the reward. This means that the $k_1$ model is generally smaller than the $k_{\text{inf}}$ model, leading to faster simulations. As shown in table 4, $k_1$ performs an average of 2360 simulations with 20ms of thinking time, while the full model only reaches 440 simulations. This increase in simulation speed results in the improved performance shown in fig. 4b.

Table 5 shows that an increase in thinking time generally increases performance, most notably when increasing from 5ms to 10ms, with diminishing returns for further increases. The differences between $k_1$ and $k_{\text{inf}}$ are also most pronounced at lower thinking times, especially with 80 additional binary factors, where the abstraction provides the most benefit. These findings demonstrate that augmenting FBA-POMCP with abstraction can increase performance through computational efficiency.

Overall, these results show that abstraction can be beneficial in multiple ways. Increasing the simulation speed leads to better performance, and simplifying the problem leads to faster learning due to greater statistical strength. The improvement in simulation speed is most pronounced when many factors are abstracted away, maximizing the difference in simulation speed between the models with and without abstraction.

### 4.3 Collision Avoidance

In the Collision Avoidance domain, the agent flies from one side to the other in a 10 (width) x 5 grid. The episode ends when the agent reaches the last column, where it has to avoid colliding with a moving obstacle. This obstacle has a 20% chance to stay stationary and otherwise randomly moves either up or down. The agent can decide to move up, down, or stay level. We increased the complexity of the original Collision Avoidance (Katt et al., 2019; Luo et al., 2019) by adding additional factors: *Speed* (slow, fast), *Traffic* (low or high amount of traffic), *Time of Day* (day, night), and *Obstacle Type* (3 types, e.g., helicopter, plane). These influence the agent's forward movement and the obstacle's behavior, fig. 5 shows the resulting dynamics. Only the obstacle's transition function is uncertain and the agent receives noisy observations of the obstacle. We evaluate the abstractions $k_0$, $k_1$, and $k_2$, which is equivalent to the full model $k_{\text{inf}}$. The abstraction $k_0$ includes the factors $x$, $y$, and *Obstacle Y*. The abstraction $k_1$ additionally includes the factors *Speed* and *Obstacle Type*, if *Obstacle Type* has a connection to *Obstacle Y*.

**Results** We again test the effectiveness of two levels of abstraction under a fixed number of simulations. One of the benefits of abstraction is a smaller and (therefore) faster model. Another benefit is an increase in statistical strength through the removal of factors. The lines in fig. 6 show the simple moving average of the return per episode, and the shaded areas show the 95% confidence interval.

We see that abstraction $k_0$ learns significantly faster than both $k_1$ and $k_{\text{inf}}$, both with a fixed number of simulations and with a fixed thinking time. The abstraction $k_0$ allows for faster learning of the transition function of *Obstacle Y* since it removes the (possible) parents *Speed* and *Obstacle Type*. When *Speed* and *Obstacle Type* are parents of *Obstacle Y* in the sampled graph structure, $k_1$ retains these connections. As a result, $k_1$ learns the transition function of *Obstacle Y* at a rate comparable to $k_{\text{inf}}$.

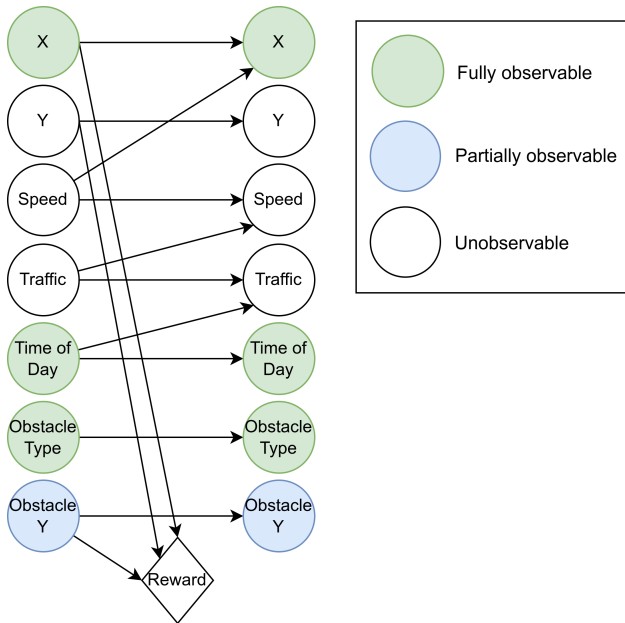

Figure 5: Ground truth graph representing the dynamics of the Collision avoidance problem.

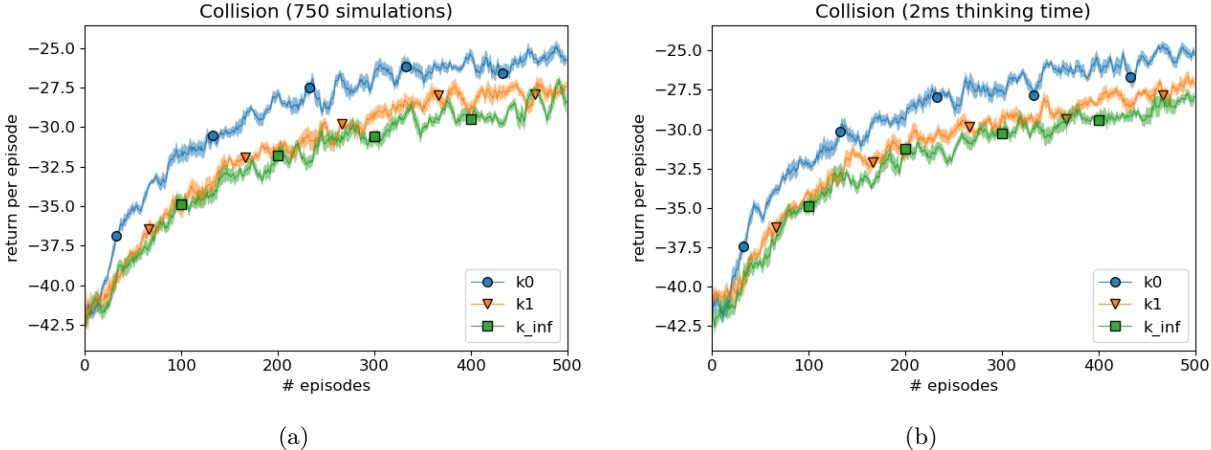

Figure 6: Performance in the Collision Avoidance domain, a) with a fixed thinking time, b) with a fixed number of simulations.

In $k_1$, *Time of Day* and *Traffic* are removed. However, since their transition functions, along with that of *Speed*, are considered known, this does lead to greater statistical strength in learning *Speed*. The impact of *Time of Day* and *Traffic* is also less pronounced, as they only influence $x$ indirectly through *Speed*. Nevertheless, $k_1$ tends to perform slightly better than $k_{\text{inf}}$. This is not due to faster simulations; in this domain, any speed up is minimal, and similar results are observed even with a fixed number of simulations. One potential advantage of removing *Time of Day* and *Traffic* is that it reduces the branching factor in the tree, as there will be no separate observations for these factors. This focuses the tree search by eliminating the need to consider these factors, albeit at a slight cost to model accuracy.

## 4.4 Room Configuration

In the Room Configuration domain, the task of the agent is to configure items in a 4 (width) x 3 grid to satisfy a teacher's preferences. Each tile contains a configurable item with two settings. The teacher is

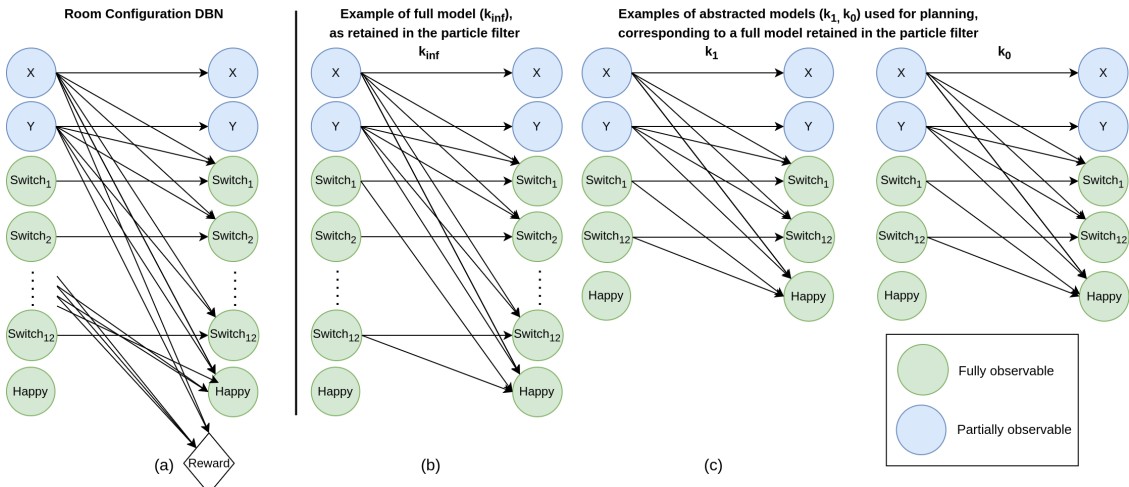

Figure 7: a) Ground truth graph representing the dynamics of the Room Configuration domain for the *switch* action, b) example of a full model in the particle filter, c) examples of two abstract models.

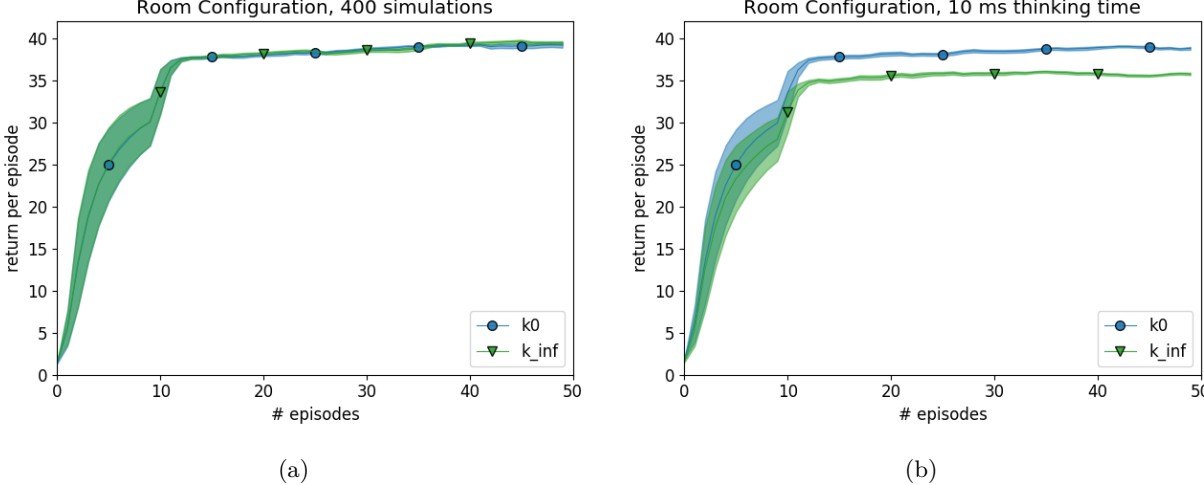

Figure 8: Performance in the Room Configuration domain, a) with a fixed thinking time, b) with a fixed number of simulations.

concerned only with the configuration of three specific items, but the agent does not initially know exactly which ones. The reward is modeled via a fully observable *Happy* factor, which takes on the same value as the reward. The agent must learn which configuration factors affect *Happy*. Small rewards or penalties are given when changing item settings, and a large reward is received once all desired configurations are set. The agent receives noisy observations of its location and has imperfect knowledge of movement success. We evaluate the abstraction $k_0$ and the full model $k_{\text{inf}}$.

**Results**   We test learning of the reward function and the advantage of abstraction when only factors that (the agent believes) are irrelevant to the reward are abstracted away. The lines in fig. 8 show the moving average of the return per episode, and the shaded areas show the 95% confidence interval. We can see that the agent learns how to perform well in 10 episodes, after which the performance remains the same.

The abstraction $k_0$ performs more simulations (455) than the full model $k_{\text{inf}}$ (365) within a fixed amount of time (10ms). This computational advantage allows $k_0$ to outperform the full model, as shown in fig. 8b.

Figure 8a further demonstrates that there is no performance difference between the two models when the number of simulations is fixed. These results show that the agent can quickly learn the structure of the reward and that abstraction can increase performance through its increased simulation speed.

## 5    Related Work

The FBA-POMDP is a factored version of the tabular BA-POMDP (Ross et al., 2011). The infinite-POMDP approach (Doshi-Velez, 2009) is a non-parametric Bayesian approach. This approach requires no knowledge of the state space. However, it assumes a hierarchical Dirichlet Process as prior, for which providing an informative prior can be difficult. BRL in continuous POMDPs typically makes Gaussian assumptions, such as in Dallaire et al. (2009), where Gaussian processes are the model of choice. To extend our approach to continuous POMDPs, DBNs that work with continuous variables (Grzegorczyk & Husmeier, 2011) could be investigated. Alternatively, constructing different levels of abstraction in a continuous domain could done through varying levels of discretization.

The BA-MDP (Duff, 2002), and the respective solution methods (Poupart et al., 2006; Ross & Pineau, 2008; Vien et al., 2013), are the fully observable counter-part to the (F)BA-POMDP. In this setting (BRL for MDPs), work has been done on exploring applications of Deep RL (Hoang & Vien, 2020; Zintgraf et al., 2021). This line of work solves the (easier) fully observable problem, and how to extend these methods to POMDPs is unclear.

Other approaches make use of recurrent networks for dealing with partial observability to generalize deep RL to POMDPs (Dung et al., 2008; Hausknecht & Stone, 2015; Zhu et al., 2017; Meng et al., 2021). However, these networks are general-purpose, requiring many samples. This realization has motivated RL-specific architectures designed to capture history efficiently (Jaderberg et al., 2019; Ma et al., 2019). Deep variational methods are another approach that can be efficient (Igl et al., 2018; Tschiatschek et al., 2018). However, none of these methods allow for encoding prior knowledge or tackling the exploration problem, to which the FBA-POMDP framework provides an elegant solution.

In the last two decades, there have additionally been many different approaches to (not deep or Bayes) learning in POMDPs, e.g., McCallum (1996); Shani et al. (2005); Azizzadenesheli et al. (2016); Liu & Zheng (2019); Bennett & Kallus (2024). However, these also do not allow for encoding prior knowledge or tackling the exploration problem.

Planning with abstractions has been studied before, mostly in the context of MDPs. The method by Dearden & Boutilier (1997), that our approach is based on, applied abstraction to factored problems. We extend this work to the partially observable BRL and planning setting and introduce a mechanism to automatically create multiple levels of abstraction based on the structure of the problem. Another line of work has focused on building abstractions during the tree search (Hostetler et al., 2014; Anand et al., 2016). Hostetler et al. (2014) also provide results for doing the tree search using an abstraction function. However, this means the simulations themselves need to be done using the full model. Chitnis et al. (2021) instead first learn an abstraction, which they then use for planning. Rather than being given an abstraction, their goal is to learn one. We investigate the effects of an abstraction method on learning efficiency and performance. He et al. (2020) do planning in the partially observable setting and construct an abstract model before the tree search. The key difference with our approach is that we do not assume access to a simulator of the real environment. Instead, we *learn* abstract dynamics while acting in the real-world.

Our work is closely related to posterior sampling approaches in causal RL, particularly the work by Mutti et al. (2024). Their method maintains a prior distribution over potential factorizations of a F-MDP and performs posterior sampling reinforcement learning accordingly. Our approach differs significantly in several aspects. First, we extend the methodology from fully observable scenarios (F-MDP) to partially observable ones (F-POMDP). Their approach explicitly notes that exact solutions for the sampled F-MDPs are required, a condition generally computationally intractable and even more prohibitive in the context of F-POMDPs. In contrast, our method circumvents this exact-solution requirement by employing an anytime planning algorithm (FBA-POMCP) within the "planning as learning" paradigm.

## 6   Discussion

A limitation of the FBA-POMCP method is that it does not plan beyond the current episode. By not planning for future episodes, it does not consider the value that knowledge obtained in the current episode can have for future episodes. Approaches that quantify information gain (Zhang et al., 2020; Ambrogioni, 2021; Liu et al., 2021) and those that plan for long horizons (Grover et al., 2020) could address this limitation, and future work could explore combining these strategies.

A difficulty of applying abstractions within FBA-POMCP lies in marginalizing the belief. As discussed in section 3.3, when aggregating counts for approximate abstractions, it is unclear what the best solution is, since this is generally unknown beforehand and can depend on the policy. Even for a fixed policy, marginalization may lead to a belief that misrepresents the true (abstract) dynamics. For example, in the corridor problem, marginalizing away the *Button* factor leads to misrepresented abstract dynamics because marginalizing it away combines the situations where the button is in the non-pushed state with those where it is in the pushed state, implicitly resulting in a belief that at each step there is a probability that the button is in the pushed state. This is a misrepresentation since the button starts in the non-pushed state and, since it is marginalized away, the agent will not learn to push the button, so it will always be in the non-pushed state. Such situations can make it more difficult for the agent to learn. An idea could be to use knowledge about the starting position and the abstracted model in the marginalization. For instance, if the *Button* factor is removed from the model, we could keep the counts for when it is in the non-pushed state and ignore the counts for the pushed state since it is always in the non-pushed state at the beginning of an episode.

In the Corridor experiment, we aimed to make the results more interpretable by showing how the agent's behavior changes during learning, across the different models. This helped clarify the sources of the observed differences in return per episode, as well as the adaptations the agent makes over time. The behavior of the smaller abstract models is easier to understand than that of the full model, and this improved interpretability offers an additional motivation for using abstraction and could be an interesting perspective for future research.

The Corridor experiment also shows that abstraction does not always help, and can in fact be detrimental. When crucial information is removed, it can make learning impossible. In general, the usefulness of abstraction depends on how much value (or information) is lost. If the loss in value is too great, more simulations will not help to achieve a better performance. On the other hand, if there is no loss in value or the loss in value is relatively small, abstraction can help improve performance by making learning easier and by speeding up simulations during planning.

Combining learning abstractions with guarantees is challenging. In fact, whether it was possible to bound the value loss of model-based RL using a given $\eta$-approximate model-similarity abstraction was an open question until recently (Starre et al., 2023). What we can say is that theoretically, for any abstraction, there exists some $\eta$ such that the bound in eq. (21) holds, even if this bound becomes vacuous at higher values However, it is challenging to estimate or verify the actual value of $\eta$ in practice. This motivates *abstraction selection* (Ortner et al., 2014; Jiang et al., 2015) as an important direction for future work, allowing agents to adaptively switch between representations over time.

Given the challenge of determining whether a specific abstraction will be beneficial, abstraction selection could be a viable approach. This can be particularly challenging in RL, where the problem is often not fully known. With abstraction selection, the algorithm would choose which abstraction to select during learning. For instance, selection could be done by deriving value loss bounds for specific abstractions (Dearden & Boutilier, 1997) and using these to make a decision. Alternatively, the problem of abstraction selection could be viewed as a non-stationary multi-armed bandit problem, where at each episode we select a (abstract) model. The problem is non-stationary since the models the agent learns change each episode with the experience it obtains, and thus the policy and the expected reward can also change. Multi-armed bandit methods that deal with such non-stationarity could be used (Garivier & Moulines, 2011; Cheung et al., 2019; Trovo et al., 2020).

Regarding the quality of the abstraction, represented by $\eta$, we can consider different quantities of a problem to determine whether $\eta$ is small. For example, in the context of approximate abstractions in MDPs, we can consider approximate model similarity or approximate $Q^*$ abstractions (Abel et al., 2016). In an approximate model similarity abstraction, the size of $\eta$ depends on how close the transition functions and rewards of the grouped states are. For approximate $Q^*$ abstractions, it depends on the maximum difference in the $Q^*$ values of the grouped states. Extending these notions to F-POMDPs requires accounting for the observation function and the factorization structure, making such extensions non-trivial. Nevertheless, existing work on approximate abstractions in MDPs provides valuable insights: results indicate that $\eta$ remains small when little information is lost through abstraction. In our setting, this suggests that $\eta$ is small when the abstraction primarily removes largely irrelevant factors.

Our abstraction approach has been developed within the framework of FBA-POMDPs. However, the proposed abstraction method is general and leverages only the structural properties of the problem, making it potentially beneficial for a broader range of F-POMDP algorithms. An interesting direction is to investigate how Thompson Sampling (Thompson, 1933), particularly its adaptation to POMDPs (Bai et al., 2014), could be combined with factored representations and our abstraction framework. Such a combination may reduce the dimensionality of the sampling space and further improve scalability.

# 7 Conclusion

We proposed combining learning and online planning for BRL for FBA-POMDPs with abstraction. We empirically showed that this combination significantly improves learning, scalability, and performance, using an intuitive and straightforward form of abstraction. This happens through several effects. First, we have shown that abstraction improves performance by increasing the simulation speed. Moreover, we have shown that abstraction improves performance even with a fixed number of simulations through greater statistical strength. With an abstract model, the agent takes different actions since it does not need to explore the dynamics of less relevant factors. Finally, the abstraction method allows FBA-POMCP to learn in very large problems with a state space up to $|\mathbb{S}| = 6 \times 10^{25}$.

To the best of our knowledge, this is the first work to explore abstraction in BRL for POMDPs, representing an initial step in investigating this combination. In the future, abstraction could also be further incorporated into FBA-POMDP by directly maintaining a belief over which factors should be part of the model.

### Acknowledgments

This project received funding from the European Research Council (ERC) under the European Union's Horizon 2020 research and innovation programme (grant agreement No. 758824 —INFLUENCE).

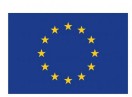 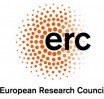

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

---

**Algorithm 5** Sequential Importance Sampling

1: **Input:** $\{\dot{s}_i, w_i\}_{i=0}^n$: current (weighted) filter
        $a, o$: action and observation
2: **for** $i \in 0, \ldots, n$ **do**
3:    $s_i' \sim p(\cdot | s_i, a; G_i, \chi_i)$
4:    $w_i' \leftarrow w_i \times p(o | s_i', a; G_i, \chi_i)$
5:    $\chi_i' \leftarrow \mathcal{U}(\chi_i, s_i, a, s_i', o)$
6: **end for**
7: **return** $\{s_i', G_i, \chi_i', w_i'\}_{i=0}^n$   // Normalize & re-sample

---

**Algorithm 6** Initialize Particle Filter

1: **Input:** $p_{\dot{s}_0}$: prior over initial state
        $n$: number of desired particles
2: **for** $i \in 0, \ldots, n$ **do**
3:    $\langle s_i, G_i, \chi_i \rangle \sim p_{\dot{s}_0}$
4:    $w_i \leftarrow \frac{1}{n}$
5: **end for**
6: **return** $\{s_i, G_i, \chi_i, w_i\}_{i=0}^n$

---

## A   Extension Belief tracking in the FBA-POMDP

The belief, the posterior over the current state, is a probability distribution over the POMDP state and its distribution $b \in \Delta(\mathbb{S} \times \mathbb{D})$ given the observed history $h_t = (a_0, o_1, \ldots, a_{t-1}, o_t)$. Unfortunately, the computation of the belief update is intractable in most problems. Thus, the belief is often approximated with particles instead (Thrun, 1999). A particle filter represents a distribution through *particles*, which in this case represent FBA-POMDP states, specifically each particle is a weighted FBA-POMDP state $(w, s, G, \chi)$ with (unnormalized) weight $w \in \mathbb{R}^+$.

There are numerous sampling mechanisms for updating the belief given a new action-observation pair (Thrun, 1999). Here, we focus on sequential importance sampling (SIS). SIS consists of two operations: propagation and re-weighting (see algorithm 5). First, the *proposal distribution* propagates a particle by sampling its next value from the FBA-POMDP transition function $p(\dot{s}' | \dot{s}, a)$. Then, the likelihood of the particle generating the received observation $p(o | \dot{s}, a, \dot{s}')$ is used to re-weight the particle (recall equation 3). The initial belief (particle filter) is initialized by sampling from the priors over the POMDP state and the DBNs describing the dynamics (see algorithm 6).

Specific to the FBA-POMDP, the belief over the topologies can deteriorate: the number of unique graph structures is determined (and limited) by the initial particle filter, as topologies do not get updated during the belief updates. When that happens, Katt et al. (2019) re-invigorate the belief with a Metropolis-Hastings-within-Gibbs sampling procedure (Murphy, 2012).

## B   Algorithms

Algorithm 7 shows the main loop of the FBA-POMCP algorithm (Katt et al., 2019). The *Simulate* that is uses is shown in algorithm 8. The GreedyActionSelection selects the action that has the highest value. In case of a tie, it randomly selects one of the actions with the highest value. Algorithm 9 shows the *Step* function when abstractions are used.

---

**Algorithm 7** FBA-POMCP

1: **Input:** $B$: particle filter with hyper-states $\dot{s}$
        num_sims: number of simulations to do.
2: $h_0 \leftarrow ()$                                              // The empty history (i.e., now)
3: **for** $i \in 1, \ldots, $ num_sims **do**
4:    // First, we root sample a hyper-state:
5:    $\dot{s} \sim B$                                                // Sample from belief
6:    Simulate$(\dot{s}, 0, h_0)$
7: **end for**
8: $a \leftarrow$ GreedyActionSelection$(h_0)$.
9: **return** $a$

---

---

**Algorithm 8** Simulate

---

1: **Input:** $\dot{s} = \langle s, G, \chi \rangle$: hyper-state
    $d$: search depth
    $h$: simulated history.
2: **if** IsTerminal$(h) || d ==$ max_depth **then**
3:  **return** 0
4: **end if**
5: $a \leftarrow$ UCTactionSelection(h)
6: $R \sim R(\dot{s}, a)$
7: $\dot{s}', o \leftarrow$ Step$(\dot{s}, a)$
8: $h' \leftarrow (h, a, o)$
9: **if** $h' \in$ Tree **then**
10:  $r \leftarrow R + \gamma$ Simulate$(\dot{s}', h')$
11: **else**
12:  ConstructNode$(h')$
13:  $r \leftarrow R + \gamma$ RollOut$(\dot{s}', h')$
14: **end if**
15: $(\ldots)$                // Update statistics in nodes
16: **return** r

---

**Algorithm 9** Step (with abstraction)

---

1: **Input:** $\bar{s}$: abstracted hyper-state
    $a$: simulated action.
2: // Recall that $\bar{s} = \langle \dot{s}, \bar{G}, \bar{\chi} \rangle$, with $\dot{s}$ containing a current state $s$.
3: $s', o \sim p_{\bar{G}, \bar{\chi}}(\cdot | s, a)$      // Sample next state and observation from abstracted counts.
4: $\dot{s}' \leftarrow \langle s', G, \chi \rangle$
5: $\bar{s}' \leftarrow \langle \dot{s}, \bar{G}, \bar{\chi} \rangle$
6: **return** $\bar{s}', o$

---

## C   Proof

We restate the Lemma from section 3.4 and give a proof sketch:

**Lemma 1.** *The result of applying the abstraction method, as described in Algorithm 2 and Section 3.3, to the original FBA-POMDP results in another FBA-POMDP, the abstract FBA-POMDP.*

*Proof.* First, we define the observation space and function, followed by the state space for different levels of abstraction. The reward function does not require changes since, during abstraction, the models in the particles always keep the factors that are believed to be part of the IR set.

As detailed in section 3.3.4, the abstract observation space $\bar{\mathbb{O}}$ is enhanced by including a "not observed" option for all observation variables. This addition addresses scenarios where the observation function depends on state factors that are abstracted away; in such cases, the observation function returns "not observed". Otherwise, the observation space and function remain unchanged.

For the state space, we distinguish abstraction level $k_0$ from other abstraction levels. For $k_0$, only the factors in the set IR are included, while higher abstraction levels can include additional factors depending on the topology.

For level $k_0$, the state space $\bar{\mathbb{S}}$ is derived by aggregating states based on the distinct values of the remaining factors. Transition functions are adjusted for the new state space through the marginalization procedure described in section 3.3.

Table 6: Fixed experiment settings.

| Parameter | Corridor | Cracky Pavement | Collision | Room Conf |
|---|---|---|---|---|
| $\gamma$ | 0.95 | 0.95 | 0.95 | 0.95 |
| # of particles in belief | 500 | 500 | 500 | 10000 |
| # of episodes | 100 | 500 | 500 | 50 |
| # of runs | 100 | 100 | 10000 | 1000 |
| Horizon ($H$) | 20 | 12 | 20 | 13 |
| UCT constant | 5 | 1 | 500 | 10 |
| Reinvigoration | Yes | No | No | Yes |
| Log-likelihood threshold | -1500 | N/A | N/A | -500 |
| State factors | 5 | $[23, 83]$ | 7 | 15 |
| $|\mathbb{S}|$ | 320 | $[5 \times 10^7, 6 \times 10^{25}]$ | 6000 | 196608 |

For higher abstraction levels, the belief specifies the factors that are relevant. For example, for abstraction level $k_1$, factors that directly influence the IR set are also considered relevant. Similar to $k_0$, factors that are never considered relevant are removed. For state factors included in only some models, the abstract state space is enhanced by adding a "not relevant" value. This value ensures transitions default to "not relevant" when the factor is not present in a model. Transitions are then adjusted for the new state space through the marginalization procedure. The observation function is also adjusted to return "not observed" for factors that return "not relevant".

Combining these transformations, we obtain a fully specified FBA-POMDP after abstraction. □

In practice, implementing the "not relevant" value is unnecessary, as only factors included in the simulated particle affect observations and actions during tree search.

# D    Extended Experiment Details

## D.1    Experimental Setup

In the experiments, we investigated various levels of abstraction across different domains and considered both a fixed amount of simulations and a fixed amount of computation time. For each level of abstraction (including the full model) and each of the different settings, we ran a separate experiment. Due to the stochasticity in the runs, we conducted up to 10000 runs for each experiment. In the figures, we report the moving average of the returns over a window of 10 ($\frac{x_n + \cdots + x_{n+9}}{10}$), with the shaded areas indicating the 95% confidence interval. To avoid cluttering the figures with markers, we placed only 5 markers per line, spaced evenly along the x-axis.

The settings of the experiments, and some specifics of the environments, are detailed in table 6. In the table, $\gamma$ denotes the discount factor, the UCT constant is the exploration constant, and the log-likelihood threshold is the threshold below which reinvigoration is triggered. The log-likelihood is obtained during the belief update, and a low log-likelihood can indicate the belief does not adequately represent the observed data. The parameters were chosen to maintain a reasonable total run time. Because of this, we ran the experiments in the Cracky Pavement Gridworld and the Collision avoidance domains without the invigoration step. We show in appendix E that the effect of abstraction is orthogonal to the effect of reinvigoration.

We performed the experiments with a fixed number of simulations on three different machines: Intel(R) Xeon(R) Gold 6130 CPU @ 2.10GHz with 384GB RAM, Intel(R) Xeon(R) Gold 5218 CPU @ 2.30GHz with 190GB RAM, and AMD EPYC 7452 32-Core Processor CPU @ 2.0GHz with 256GB RAM. For the experiments with a fixed amount of computation time, we used (2 cores of) an AMD EPYC 7452 32-Core Processor CPU @ 1.5GHz with 512GB RAM. The software is written in C++.

### D.2 Domain Descriptions

#### D.2.1 Corridor

The Corridor domain is the domain described in the running example in section 2.1, where there are 8 different *Persons* that can be present. Each person has a slightly different probability of opening the door, such that the probability that the person opens the door during an episode is approximately between 0.0125 and 0.1. In this domain, we assume prior knowledge of all observation functions and the locations of the start, boots, button, door, and goal. In addition, we assume the structure of the transition functions of the factors *Button*, *Boots* and *Person* are known, in other words, the prior includes only models where the parents of these factors are correctly specified.

What is unknown is the structure of the transition function of the $x$-position, for the actions left and right. For this transition, the prior includes the following combinations of factors as parents for the $x$-position transitions, 1) $x$, 2) $x$ and *Boots*, 3) $x$, *Boots*, and *Door*, and 4) $x$, *Boots*, *Door*, and *Button*. Each of these combinations is assigned the same probability in the prior. The transition function of the door is also not fully known, with the prior specifying a 50% chance of *Button* being present as a parent. The prior is initialized optimistically, and so the agent initially overestimates its chances of success and the probability that the person that is present will open the door. It has to learn the correct structure and transition probabilities.

We consider the abstractions $k_0$, $k_1$, and the full model $k_{\text{inf}}$. The abstraction $k_0$ only includes the factor $x$. Abstraction $k_1$ includes $x$ and, depending on the structures included in the belief, can also include the factors *Boots*, *Button* and *Door*.

#### D.2.2 Cracky Pavement Gridworld

The Cracky Pavement Gridworld is a grid world as shown in fig. 3a. The DBN of the domain is shown in fig. 3b. The state space is factored into the $x$ and $y$ locations, *Rain*, and several extra binary factors. These extra factors could be global (e.g., light conditions) or local (e.g., presence of a chair in a specific location). In reality, only $x$, $y$, and *Rain* influence the movement of the agent, as can be seen by the incoming edges of $x$ and $y$. The "Trap" and "Vendor" tiles depicted in fig. 3 do influence the movement of the agent but are not included as separate factors because their dynamics are already captured by $x$, $y$, and *Rain*. Their interaction is described in the next paragraph.

The agent is initially located at "Start" and is running low on battery, so it has to move to a charging station at one of the "Goal" locations. The agent only observes its $x$ and $y$ location and does so with a noisy sensor. For both $x$ and $y$, the agent makes the correct observation 90% of the time. If an incorrect observation occurs, a randomly selected adjacent location is returned. At the edges, the probability of receiving the correct observation increases to 95%. The agent can move in all four directions, but the movements can fail. The chances of movement failure are influenced by a global factor called *Rain* which represents whether the tiles are dry or wet. The *Rain* factor is initialized randomly, and every timestep there is a 5% chance that the rain condition changes. On normal tiles, actions succeed 95% of the time when there is no rain and 66% of the time when it is raining. However, on the Trap tiles with cracked pavement, moves succeed only 10% of the time. Additionally, when it is not raining, a vendor with a cart occupies the Vendor tile, making it harder to move past with only a 10% success rate for moving. Conversely, during rain, the Vendor tile is vacant and functions like a normal tile. Therefore, to behave optimally, the agent should traverse the Vendor tile when it is raining and circumvent it when dry.

In this domain, we assume prior knowledge of all observation functions, the transition functions of *Rain* and the extra binary factors, and the start and goal locations. Additionally, we assume that it is known that the $x$ and $y$ factors both (at least) depend on each other. However, there is no prior knowledge of the trap locations and the vendor locations, meaning that none of the possible count tables in the belief space specify different movement probabilities on these locations. Some of the graph structures in the belief space include *Rain* and/or extra binary factors (three in fig. 3b) as parents of $x$ and $y$, implying the agent does not know whether or not these factors influence its movement. The extra binary factors are initialized randomly and have a 20% of changing at each step. In this problem, the agent has to learn that $x$, $y$, and *Rain* are the

only relevant factors for its movement. This task is complicated by the interaction between the trap states and several uninformative factors, as the agent may mistakenly attribute its inability to move on trap states to the presence of some of these uninformative factors.

As shown in table 6 on page 30, we run the experiments with two different amounts of extra binary factors: 20 and 80. The total amount of state factors is 23 and 83, respectively, since both settings also have the $X, Y$, and $Rain$ factors. Scalability in the number of factors is very hard and important because the size of the state space, and the possible graph structures, grow exponentially with the number of factors. With 20 and 80 extra binary factors this domain has a state space of approximately $5 \times 10^7$ and $6 \times 10^{25}$ states, respectively. Factored representations are needed to find solutions for problems of such sizes. Flat learning methods like Bayes-Adaptive POMCP (Katt et al., 2017) are not feasible here, even in the simple case of 20 extra binary factors. This is because representing the transition table for just 1 action, a table of size $|S|^2$, requires more than 9 million GB per particle.

We consider the abstractions $k_0$, $k_1$, and the full model $k_{\mathrm{inf}}$. The abstraction $k_0$ includes only the factors $x$ and $y$ since they are the ones that directly influence the reward (fig. 3b). Abstraction $k_1$ also includes the parents of $x$ and $y$. We scale the number of extra binary factors to test the speedup and to see how it affects the performance.

### D.2.3 Collision Avoidance

In the Collision Avoidance domain, the agent flies from one side to the other in a 10 (width) x 5 grid. The episode ends when the agent reaches the last column, where it has to avoid colliding with a moving obstacle. This obstacle has a 20% chance to stay stationary and otherwise randomly moves either up or down. The agent can decide to move up, down, or stay level.

We increased the complexity of the original Collision Avoidance (Katt et al., 2019; Luo et al., 2019) by adding additional factors. These additional factors influence those in the original problem. Figure 5 shows the resulting dynamics. We added the factors *Speed* (slow, fast), *Traffic* (low or high amount of traffic), *Time of Day* (day, night), and *Obstacle Type* (3 types, e.g., helicopter, plane). *Obstacle Type* and *Time of Day* are fully observable and do not change during the episode. The agent receives a noisy observation of the obstacle (accurate around 80% of the time). The agent has an 85% chance to move one cell forward. If the *Speed* is high, it has a 15% chance to move forward two columns. If the *Speed* is low, it has a 15% chance to stay in the same column.

The *Speed* is influenced by the *Traffic*. When the amount of *Traffic* is low, there is a 90% chance that the *Speed* changes to (or stay at) high and a 10% chance that the *Speed* changes to (or stay at) low. This is reversed when the *Traffic* is high. The *Traffic* is influenced by the *Time of Day* in a similar way, when it day there is a 90% chance that the *Traffic* changes to (or stay at) high and a 10% chance that the *Traffic* changes to (or stay at) low. This is reversed when it is night. The Time of Day is randomly chosen at the start of an episode, with a 50% chance of either day or night.

We assume prior knowledge of all transition and observation functions, except for the transition probabilities of the obstacle. There are four different graph structures for the obstacle transition function to which the prior assigns a positive probability. These structures include: 1) the structure where *Obstacle Y* is only influenced by itself; 2 and 3) the structures where it is influenced by itself and either *Speed* or *Obstacle Type*; and 4) the structure where it is influenced by itself, *Speed*, and *Obstacle Type*. Each structure has a 25% probability.

We consider the abstractions $k_0$, $k_1$, and $k_2$, which is equivalent to the full model $k_{\mathrm{inf}}$. The abstraction $k_0$ includes the factors $x$, $y$, and *Obstacle Y*. The abstraction $k_1$ additionally includes the factors *Speed* and *Obstacle Type*, if *Obstacle Type* has a connection to *Obstacle Y*.

### D.2.4 Room Configuration

In the Room Configuration domain, the task of the agent is to set up a classroom in a desirable way for a teacher. We model this environment as a 4 (width) x 3 grid, where each tile contains a configurable item

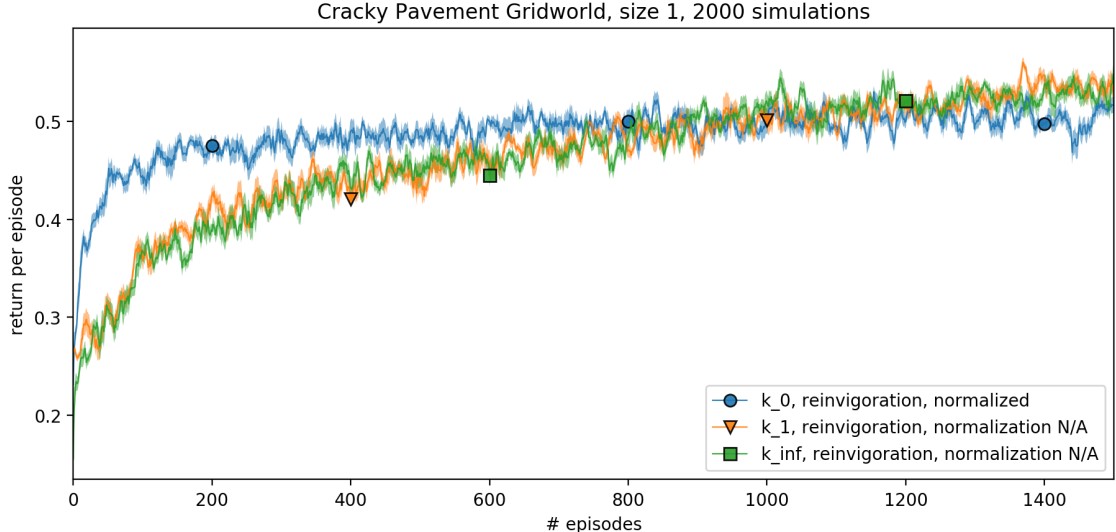

Figure 9: Comparison with invigoration for a longer number of episodes.

with two settings. The agent can change the configuration of these items. The teacher is concerned only with the configuration of three specific items and is happy once these are configured correctly.

The reward function is (largely) unknown to the agent in this domain, as the agent does not initially know configuration factors that influence the reward. To address this, we model the reward with a state factor called *Happy*, which takes on the same value as the reward and is fully observable. While *Happy* is fully observable, the agent does not know exactly which configuration factors influence it. That is, the prior belief assigns a non-zero probability to multiple sets of parents of *Happy*. Therefore, the agent must learn which configuration factors are relevant to the reward. The *Happy* factor has four different states: neutral (0 reward), slightly unhappy $(-1)$, slightly happy $(+1)$, and very happy $(100)$. The agent receives a reward when it performs the switch action, and this reward depends on its location and the status of the configurable items. Specifically, the agent receives a small reward or penalty $(\pm 1)$ when it changes the configuration of an item to the correct or incorrect setting, respectively. Once it sets the configuration of all three items the teacher desires, it receives a large reward $(100)$.

While the configurations of the items are fully observable, the agent receives noisy observations of its own location. For both $x$ and $y$, the agent makes the correct observation $95\%$ of the time. If an incorrect observation occurs, a randomly selected adjacent location is returned. At the edges, the probability of receiving the correct observation increases to $97.5\%$. Movement actions have a $95\%$ chance of moving in the intended direction, whereas the *switch* action to change the item settings is always successful. The agent has prior knowledge of the observation function and the transition function for switching. Additionally, all the possible initial belief states underestimate the probability that the move action is successful.

We consider the abstraction $k_0$ and the full model $k_{\inf}$. The abstraction $k_0$ keeps the factors connected to the reward. The structures in the prior belief always include $x, y$ as parents of *Happy*. The prior belief is defined such that it assigns a non-zero probability only to structures where the abstract models $k_1$ and $k_0$ are identical. Thus, in the experiment, we only use $k_0$.

# E   Further Experiments

Here, we demonstrate that abstraction can improve performance when we use the invigoration step from FBA-POMCP, and that $k_1$ and the full model $k_{\inf}$ can eventually outperform $k_0$. We conducted 100 runs for each setting and report the simple moving average of the return per episode, with the shaded areas showing the standard error.

We ran an experiment with invigoration over a larger number of episodes, as shown in fig. 9. Initially, $k_0$ outperforms $k_1$ and $k_{\text{inf}}$. However, with enough data, the $k_1$ and $k_{\text{inf}}$ models become accurate enough to surpass $k_0$ in performance. The $k_1$ model matches the performance of $k_{\text{inf}}$ because the number of simulations is fixed.

For larger problem sizes and with a fixed thinking time instead of a fixed number of simulations, $k_1$ is expected to initially outperform $k_{\text{inf}}$. Additionally, $k_0$ should show an even greater initial improvement over $k_{\text{inf}}$, and it could take even longer for $k_{\text{inf}}$ to surpass the performance of $k_0$.

