# OpenReview forum: "Abstraction for Bayesian Reinforcement Learning in Factored POMDPs"
_TMLR — Accepted by TMLR_

### Review · Reviewer_9cu2 · 2025-02-24

**Summary Of Contributions:**

This paper aims at scaling Bayesian RL to (possibly large) factored POMDPs through abstractions, which filter out factors that do not have direct dependencies with the reward.  Coarsely, the abstractions work as follows: First are considered only the features that affect the reward directly; then, the parents of these features, and so on and so for to obtain different degrees of abstractions. The paper provides a small theoretical corroboration and an empirical evaluation in illustrative, discrete domains to show that: Abstractions reduce the size of the model; Abstractions improves the overall performance by speeding up learning.

**Audience:**

Yes

**Broader Impact Concerns:**

the paper can be categorized as fundamental research. The broader impact of the results can be hardly assessed at this stage of research.

**Claims And Evidence:**

Yes

**Requested Changes:**

Instead of requesting changes, I am using this space to report some questions I had while reading (which may be addressed in a revised manuscript or in the discussion) and some suggestions on the presentation.

Questions:
1) The abstractions are here proposed for Bayesian-adaptive approach to Factored POMDPs. However, the way abstractions are constructed is general. Can they provide benefit to other algorithms for Factored POMDPs? An interesting instance is Thompson sampling.
2) The way abstractions are built, the increasing degree of abstraction allows to consider factors that affect the reward less directly. Long-term dependencies could escape the short chain of dependencies considered with smaller $k$. An approach to target this could be to model the factors dependent with the value of a policy instead of the reward. However, this can easily increase the complexity of the model.
3) Developing Proposition 2 further, can we say anything about when $\eta$ is expected to be small?

Suggestions: I am providing here below some suggestions on how the presentation of the paper may be tweaked. Clearly, those are mostly a matter of taste and they are not critical for my recommendation. I am reporting them anyway in case they can be useful for future versions of the manuscript.

The paper feels a little bit unfocused and over-detailed, e.g., it takes almost 8 pages before the way abstractions are built is described and only after the discussion it was clear to me that the aim of the paper is to extend the work of Dearden & Boutillier to POMDPs (abstractions are essentially similar to theirs, aside from partial observability issues). Instead, a lot of space is dedicated to present the background (which is non-standard to be fair) and various details that shall mostly belong to the appendix (such as machines used for computation and extended empirical details).

Perhaps, the authors may consider restructuring the paper in these terms: Presenting the idea as an extension of the work by Dearden & Boutillier to Factored POMDPs (in general, without tying it to Bayes-adaptive methods) and focusing the text on abstractions and additional challenges of partial observability, possibly leaving details of the setting to prior works. I do not see particular reasons why this paper could not fit into a "short submission" (12 pages by TMLR standard) more centered on the core contribution rather than the general background.

Another aspect that I found interesting but not fully developed is the interpretability of the results. It is very nice to be able to go through the experiments and understand how the policy is changing while learning. Perhaps interpretability could also be mentioned as a motivation for abstractions.

Finally, some works on *causal RL*, where causal factors on the transitions dynamics and/or reward are studied, could be mentioned in the manuscript. Some pointers are:
- Zhang et al. 2020, Invariant causal prediction for block mdps;
- Tomar et al. 2021, Model-invariant state abstractions for model-based reinforcement learning;
- Gasse et al. 2021, Causal reinforcement learning using observational and interventional data;
- Feng et al. 2022, Factored adaptation for non-stationary reinforcement learning;
- Mutti et al. 2023, Provably efficient causal model-based rl for systematic generalization;
- Mutti et al. 2024, Exploiting causal graph priors with posterior sampling for reinforcement learning.

The latter, especially, deals with extending Thompson sampling for priors over factors topology for Factored MDPs, which may have some interesting connections with this paper.

**Strengths And Weaknesses:**

Strengths:
- The paper is well-written and very detailed (sometimes even too detailed, more on that below);
- The paper tackles a general framework of Bayesian RL for Factored MDPs, which technically subsumes a wide range of domains and learning algorithms;
- The paper provides various interpretable empirical results to support the proposed abstractions.

Weaknesses:
- The writing is very extended and not clearly focused on the core contributions of the paper;
- While the framework is general, it is known to be impractical in many practical scenarios. The method here is indeed striving to make the approach more practical, but the experiments are bound to synthetic domains with categorical factors;
- The theoretical corroboration is nice but leaves out the most interesting question: How abstraction affects optimality (aka the $\eta$ value in the paper).

---

> ### Author Response · Authors · 2025-05-28
> **Author response to review by Reviewer 9cu2, part 1**
>
> Dear reviewer,
>
> Thank you for the thoughtful feedback and interesting questions. We will address the questions and comments below.
>
> “*A lot of space is dedicated to present the background (which is non-standard to be fair) and various details that shall mostly belong to the appendix (such as machines used for computation and extended empirical details).*”
> We have moved some details of the background and parts of the descriptions and details of the experimental section to the Appendix.
>
> “*The abstractions are here proposed for Bayesian-adaptive approach to Factored POMDPs. However, the way abstractions are constructed is general. Can they provide benefit to other algorithms for Factored POMDPs? An interesting instance is Thompson sampling.*”
>
> We agree and this is an interesting point, we have included the following in the discussion:
> ‘Our abstraction approach has been developed within the framework of FBAPOMDPs. However, the proposed abstraction method is general and leverages only the structural properties of the problem, making it potentially beneficial for a broader range of FPOMDP algorithms. An interesting direction is to investigate how Thompson Sampling (thompson1933likelihood), particularly its adaptation to POMDPs (bai2014thompson), could be combined with factored representations and our abstraction framework. Such a combination may reduce the dimensionality of the sampling space and further improve scalability.’
>
> Thompson, W. R. (1933). On the likelihood that one unknown probability exceeds another in view of the evidence of two samples. Biometrika, 25(3/4), 285-294.
> Bai, A., Wu, F., Zhang, Z., & Chen, X. (2014, May). Thompson sampling based Monte-Carlo planning in POMDPs. In Proceedings of the International Conference on Automated Planning and Scheduling (Vol. 24, pp. 29-37).
>
> “*The way abstractions are built, the increasing degree of abstraction allows to consider factors that affect the reward less directly. Long-term dependencies could escape the short chain of dependencies considered with smaller k. An approach to target this could be to model the factors dependent with the value of a policy instead of the reward. However, this can easily increase the complexity of the model.*”
>
> This is an interesting idea, and we agree that capturing long-term dependencies is important. In our view, increasing the value of $k$ serves a similar purpose as it allows the model to account for longer-term effects.
> We are, however, not quite sure what it means to “model the factors dependent with the value of a policy instead of the reward”, could you clarify?
>
> “*Developing Proposition 2 further, can we say anything about when η is expected to be small?*”
>
> We agree this is an interesting question that deserves more attention. We have included a small background section on abstraction (2.4) and refer back to this and $\eta$ in Section 3.4 in the paragraph following Definition 2. Additionally, we have also included the following in the discussion section:
> ‘Regarding the quality of the abstraction, represented by $\eta$, we can consider different quantities of a problem to determine whether $\eta$ is small. For example, in the context of approximate abstractions in MDPs, we can consider approximate model similarity or approximate $Q^\*$ abstractions (Abel et al., 2016). In an approximate model similarity abstraction, the size of $\eta$ depends on how close the transition functions and rewards of the grouped states are. For approximate $Q^\*$ abstractions, it depends on the maximum difference in the $Q^\*$ values of the grouped states. Extending these notions to FPOMDPs requires accounting for the observation function and the factorization structure, making such extensions non-trivial. Nevertheless, existing work on approximate abstractions in MDPs provides valuable insights: results indicate that $\eta$ remains small when little information is lost through abstraction. In our setting, this suggests that $\eta$ is small when the abstraction primarily removes largely irrelevant factors.’
>
> Abel, D., Hershkowitz, D., & Littman, M. (2016, June). Near optimal behavior via approximate state abstraction. In International Conference on Machine Learning (pp. 2915-2923). PMLR.

---

> ### Author Response · Authors · 2025-05-28
> **Author response to review by Reviewer 9cu2, part 2**
>
> Regarding the suggestion to present the idea as an extension of the work by Dearden & Boutillier to Factored POMDPs without tying it to Bayes-adaptive methods.
> Important to note that the work by Dearden & Boutillier deals with the dynamic programming setting, where the MDP is fully known. The goal there is to take a large but fully-known MDP and reduce its size to make solving it easier. We have updated the end of the introduction (paragraphs 5-7) to reflect that we believe the most important novelty of our work over theirs is going from the planning setting to the Bayesian reinforcement learning setting, where the underlying POMDP dynamics (i.e. transition and observation) are not known beforehand. This introduces the challenges, such as the abstraction needing to work well for learning, and being able to adapt the abstract model to what is being learned. The Bayes-adaptive methods are important for efficient learning, since they cover exploration and the ability to incorporate prior knowledge.
>
> "*Perhaps interpretability could also be mentioned as a motivation for abstractions*"
>
> That is indeed an interesting perspective, and we have added a small paragraph in the discussion along these lines:
> “In the Corridor experiment, we aimed to make the results more interpretable by showing how the agent's behavior changes during learning, across the different models. This helped clarify the sources of the observed differences in return per episode, as well as the adaptations the agent makes over time. The behavior of the smaller abstract models is easier to understand than that of the full model, and this improved interpretability offers an additional motivation for using abstraction and could be an interesting perspective for future research.”
>
> Regarding pointers to causal RL.
> We thank the reviewer for the pointers. It seems Mutti et al. 2024 is especially relevant for our work. They seem to maintain a prior over possible factorizations for a factored MDP and then doing posterior sampling reinforcement learning. Our work distinguishes from theirs in several critical aspects. Firstly, we move from full to partial observability (i.e. from FMDP to F-POMDP). As they state, their approach requires the F-MDPs to be solved exactly during posterior sampling, which is in general intractable. This intractability is even a bigger issue in F-POMDPs. Our approach does not require solving the F-POMDP exactly, and instead relies of an anytime planning algorithm (i.e. BA-POMCP) within the “planning as learning” framework. We have added a discussion on this paper in the last paragraph of our related works section.

---

### Review · Reviewer_q3iN · 2025-02-26

**Summary Of Contributions:**

The authors study abstraction in POMDP.
The paper is highly cluttered, making it hard to understand the contribution.

**Audience:**

Yes

**Broader Impact Concerns:**

I defer the answer for later.

**Claims And Evidence:**

No

**Requested Changes:**

Please make the paper concrete and concise, with clear definition of all the terminologies used.

**Strengths And Weaknesses:**

While abstraction is an essential subject in RL, I genuinely have a hard time following the paper. I only managed to read up to page 10.

The paper is highly cluttered and hard to follow and understand. The authors are highly encouraged to rewrite the paper with a well-defined flow and well-defined, well-defined notation and concepts. Please mathematically define everything in a concise manner, including the concept of abstraction.

Please also have a senior person read the paper, to avoid cluttered language, e.g.,
"After retrieving a subset of state factors, we have to construct the abstract model." no, you don't "have to."
"We explore 2 first ideas." what "2" first ideas? From what ordered set of ideas? Why 2, not two?
title of 3.3.1,
"Before considering the case of CCTs, we treat the case of probability distributions." please use the terms carefully without clutter.
The use of "will" in many places.
...

The first time the abbreviation BA-POMCP is used, it is not defined.

"in section 2.2, which will turn out to be a (belief-space) Partially-observable Markov decision process itself." This has been known for a very long time.

The authors have already defined the POMPD abbreviation; there is no need to have it redefined in 2.1.

In line 36 of page 3, there is an x on the left-hand side but no x on the right-hand side.

Up to the place I have read, the paper relies on the graphical model, and there is no use or relevance to DBNs. Why bring DBN in?

In the line above eq 1, why do we have the mentioned equality for the R? Is there no uncertainty about the reward function? Please make sure it is clarified in the draft.

In the second line of "Action selection in the FBA-POMDP," the authors talk about high-dimensional problems. That is not even the setting of this work; there is no dimension here; what do the authors mean?

"The simulation traverses to a leaf by picking actions according to UCB." How is it done?? The paper mentioned is for bandits.

Does the agent have access to counts? How? Isn't it POMDP?

Who provides the abstractions? Are these given? If so, that is a very strong assumption.

In Eq 4, I don't follow the equality. Please distinguish between random variables and their realization. What is that summation over? Collection of random variables Y?

---

> ### Author Response · Authors · 2025-05-28
> **Author response to review by Reviewer q3iN**
>
> Dear reviewer,
>
> Thank you for the suggestions. We have updated the paper to address some comments and add further clarifications.
>
> Regarding some specific questions:
> We would like to clarify that the graphical model we refer to is indeed a Dynamic Bayesian Network (DBN)—specifically, a two-stage DBN as described by Boutilier et al. We have updated the text in the “Factorization” paragraph in section 2.1 to make this clearer.
>
> Regarding the agent’s access to counts: these are not directly observable, as they are part of the hidden state. However, the Bayes-Adaptive POMDP (BA-POMDP) framework allows us to treat this as a standard POMDP by maintaining a belief over the latent counts. We clarify this point in the first paragraph (and footnote 2) in Section 2.2.
>
> Finally, the abstractions used in our approach are not assumed to be given. Instead, our method automatically constructs them based on the connectivity of the factors to the reward (and the abstraction level “k”). We added a remark to clarify this earlier (in the penultimate paragraph of the introduction). To better introduce the concept of abstraction, we added a background section 2.4.

---

### Review · Reviewer_5AP2 · 2025-05-04

**Summary Of Contributions:**

This paper approaches the POMDP problem by combining factored state abstractions and Bayesian RL. They do so by representing the problem using a factor graph (or DBN); parts of the DBN are known and others are unknown. Over time, the agent updates its beliefs about the unknown parts of the DBN representation of the problem. The factored model is then used by a POMCP-style planner to obtain a belief conditioned policy. Experimental results ablate the importance of abstraction towards solving large POMDPs.

**Audience:**

Yes

**Claims And Evidence:**

Yes

**Requested Changes:**

1. Be more upfront about what parts of the POMDP are assumed to be given in the Background and Methods section.
2. Consider adding common POMDP baselines.
3. Improve the theoretical claims made in Section 3.4

For more details on these changes, please refer to the "weaknesses" portion of my review.

**Strengths And Weaknesses:**

Strengths:
* This paper is impressively ambitious---it attempts to solve the very challenging problem of online planning in large POMDPs, all while learning factored abstractions suitable for Bayesian RL.
* The paper is very well written: even though I am not super familiar with Bayesian RL, the background section gave me enough context to follow along. Similarly, although I am not familiar with FBA-POMDPs, I enjoyed learning about them.
* The problem of marginalizing certain variables in the abstract DBN was well motivated and well explained.
* The experimental section was thorough: results, error bars, evaluation metrics and computational resources were all clearly expressed. Furthermore, the trade-off inherent in using more abstract models was well studied. Furthermore, they successfully solved moderately large POMDPs using the proposed techniques.

Weaknesses:
* The biggest weakness was that I discovered more assumptions as I kept reading the paper. For e.g., from the background section, I was expecting the DBN and the CPTs to all be learned autonomously, but different experiments used varying degrees of domain knowledge. Although this was explicit in the experiments section, a brief note explaining what parts of the POMDP are assumed vs learned in the main text would be helpful.
* There were no other POMDP baselines. What about PBVI? Particle filtering + RL? Having some common baselines would further improve the experimental section.
* Some of the assumptions made by the algorithm seemed quite extreme: for e.g., "Assumption 1" and the normalization factor in Equation 15, which if I am not mistaken, would not only require domain knowledge about which variables constitute the state, but also how many possible values they can take on (also doesn't apply to continuous domains).
* The discussion in Sec 3.4 seemed superfluous. Unless I missed something, the authors simply state that if "good" abstractions are given to the agent, it can achieve bounded value error. But does the proposed algorithm discover such an abstraction? What sort of state abstraction does it learn (in the taxonomy of Abel et al which the authors cite)? How do we know that the value function learned with the learned abstractions have bounded error?

Despite these weaknesses, I enjoyed reading this paper and believe that it is an interesting approach towards doing RL in POMDPs.

---

> ### Author Response · Authors · 2025-05-28
> **Author response to review by Reviewer 5AP2, part 1**
>
> Dear reviewer,
>
> Thank you for the kind words and the feedback.
>
> Regarding the comments,
>
> “*Be more upfront about what parts of the POMDP are assumed to be given in the Background and Methods section.*”
>
> We have revised the text in the paragraph Factorization in 2.1 and in the paragraph Belief tracking in the FBA-POMDP in 2.3 to clarify what elements of the POMDP are assumed versus learned.
> Regarding the specific points:
>
> “*I was expecting the DBN and the CPTs to all be learned autonomously, but different experiments used varying degrees of domain knowledge. Although this was explicit in the experiments section, a brief note explaining what parts of the POMDP are assumed vs learned in the main text would be helpful.*”
>
> We have now added a clarifying note in the text in the paragraph Factorization in 2.1 and in the paragraph Belief tracking in the FBA-POMDP in 2.3 to clarify which components are assumed and which are learned in our approach.
>
> “*Some of the assumptions made by the algorithm seemed quite extreme: for e.g., "Assumption 1" and the normalization factor in Equation 15, which if I am not mistaken, would not only require domain knowledge about which variables constitute the state, but also how many possible values they can take on (also doesn't apply to continuous domains).*”
>
> The assumption 1 is used only within Section 3.3.3 to highlight the distinction between cases where this assumption holds and where it does not, which we have now highlighted in the first paragraph of section 3.3.2. We emphasize that the more interesting case is when it does not hold. In Section 3.4 and our experiments, we do not assume that Assumption 1 holds.
>
> As for the normalization factor, it is correct that it requires domain knowledge, and we acknowledge that this is indeed a limitation. We generally assume that the state space is known, including which variables it includes and the possible values they can take, though some of these variables may be irrelevant to the model. We have clarified that our focus is on discrete POMDPs in the fifth paragraph of the introduction, and we explicitly state the known state space assumption in the paragraph titled Factorization in Section 2.1.
>
>
> “*Consider adding common POMDP baselines.*”
>
> Thank you for the suggestion. The goal of our paper is to demonstrate how abstraction can improve the performance of Bayesian reinforcement learning in POMDPs via the BA-POMDP model. This is why our experiments focus on comparisons between the cases of abstraction and no abstraction. In this context, the baseline is the “k_inf” result, representing the performance of the FBA-POMCP algorithm without abstraction. While it may seem reasonable to consider comparisons to other POMDP solvers such as PBVI, since the BA-POMDP is itself a POMDP, such comparisons are not directly applicable. Methods like PBVI require exact belief updates, which are infeasible in the BA-POMDP setting (unless one converts it into a finite POMDP, which is impractical). We have updated the first paragraph in section 2.3 to clarify this.
>
> We are not completely certain what the reviewer means with a PF+RL approach, but our approach indeed can be interpreted as such.

---

> ### Author Response · Authors · 2025-05-28
> **Author response to review by Reviewer 5AP2, part 2**
>
> “*Improve the theoretical claims made in Section 3.4” and “The discussion in Sec 3.4 seemed superfluous. Unless I missed something, the authors simply state that if ‘good’ abstractions are given to the agent, it can achieve bounded value error.*”
>
> Section 3.4 is intended to clarify the conceptual link between abstraction and the performance of FBA-POMCP, rather than to provide strong theoretical guarantees. The results are primarily for defining relationships mathematically, aiming to illustrate how abstraction error can affect value estimation. We believe it is still useful to show that the method is theoretically sound and can perform well given sufficiently good, though not necessarily exact, abstractions.
>
> “*But does the proposed algorithm discover such an abstraction?*”
>
> Our approach, via Bayesian learning, does discover abstractions that match the observed history, from within a particular class of abstractions defined by ‘k’. It is impossible to guarantee that there will be a good abstraction in that class: For any abstraction within the class defined by ‘k’, there is an associated abstraction level $\eta$. However, we cannot guarantee a priori that this level will be low, as it depends on both the environment’s structure and the choice of ‘k’. As demonstrated in the experiments, the quality of the resulting abstraction (and thus the performance) can vary considerably with the abstraction level.
>
> “*What sort of state abstraction does it learn (in the taxonomy of Abel et al, which the authors cite)?*”
>
>  Our method does not target a specific type of state abstraction within Abel et al.’s taxonomy. Abel’s taxonomy describes different ‘types’ of abstraction in terms of how well they can represent the true dynamic, the true value function, or the true optimal policy, for instance. By doing Bayesian learning over what the dynamics function is, our approach is most aligned with learning a “model irrelevance abstraction”, now also included in the background. However, we point out that Abel’s definitions are for MDPs, while we are working with POMDPs, such that not everything neatly carries over. That said, the abstractions that we find may implicitly conform to one or more of these types at a particular abstraction level and error $\eta$.
>
> “*How do we know that the value function learned with the learned abstractions have bounded error?*”
>
> This is an important question, and we have added the following to the discussion:
> ‘Combining learning abstraction with guarantees is challenging. In fact, whether it was possible to bound the value loss of model-based RL using a given epsilon-model similarity abstraction was an open question until recently (Starre2023TMLR). What we can say is that theoretically, for any abstraction, there exists some $\eta$ such that the bound in Equation X holds, even if this bound becomes vacuous at higher values. However, it is challenging to estimate or verify the actual value of $\eta$ in practice. This motivates abstraction selection as an important direction for future work, allowing agents to adaptively switch between representations over time.’
>
> We found these to be insightful and important questions, and have further expanded the discussion section:
>
> ‘Regarding the quality of the abstraction, represented by $\eta$, we can consider different quantities of a problem to determine whether $\eta$ is small.
> For example, in the context of approximate abstractions in MDPs, we can consider approximate model similarity or approximate $Q^\*$ abstractions (abel2016near).
> In an approximate model similarity abstraction, the size of $\eta$ depends on how close the transition functions and rewards of the grouped states are.
> For approximate $Q^\*$ abstractions, it depends on the maximum difference in the $Q^\*$ values of the grouped states.
> Extending these notions to FPOMDPs requires accounting for the observation function and the factorization structure, making such extensions non-trivial.
> Nevertheless, existing work on approximate abstractions in MDPs provides valuable insights: results indicate that $\eta$ remains small when little information is lost through abstraction.
> In our setting, this suggests that $\eta$ is small when the abstraction primarily removes largely irrelevant factors.’

---

### Decision · Action_Editor_ju3x · 2025-06-12

**Recommendation:** Accept as is

**Additional Comments:**

The final comments from the reviewers are to consider still sharpening the conciseness of the writing. This is not required for the final submission, but you do have one more chance for edits towards this goal.

**Audience:**

Yes

**Audience Explanation:**

As pointed out by the reviewers, this paper should be of interest to many in the POMDP community.

**Claims And Evidence:**

Yes

**Claims Explanation:**

The paper clearly lays out related work, justifying claims about the novelty of the approach. The experiments were thorough and rigorous, with well thought-out ablations and careful analysis.